# Personalized visual encoding model construction with small data

Zijin Gu [1], Keith Jamison[2], Mert Sabuncu[1,2] & Amy Kuceyeski [2✉]

Quantifying population heterogeneity in brain stimuli-response mapping may allow insight into variability in bottom-up neural systems that can in turn be related to individual's behavior or pathological state. Encoding models that predict brain responses to stimuli are one way to capture this relationship. However, they generally need a large amount of fMRI data to achieve optimal accuracy. Here, we propose an ensemble approach to create encoding models for novel individuals with relatively little data by modeling each subject's predicted response vector as a linear combination of the other subjects' predicted response vectors. We show that these ensemble encoding models trained with hundreds of image-response pairs, achieve accuracy not different from models trained on 20,000 image-response pairs. Importantly, the ensemble encoding models preserve patterns of inter-individual differences in the image-response relationship. We also show the proposed approach is robust against domain shift by validating on data with a different scanner and experimental setup. Additionally, we show that the ensemble encoding models are able to discover the inter-individual differences in various face areas' responses to images of animal vs human faces using a recently developed NeuroGen framework. Our approach shows the potential to use existing densely-sampled data, i.e. large amounts of data collected from a single individual, to efficiently create accurate, personalized encoding models and, subsequently, personalized optimal synthetic images for new individuals scanned under different experimental conditions.

[1] School of Electrical and Computer Engineering, Cornell University, Ithaca, NY, USA. [2] Department of Radiology, Weill Cornell Medicine, New York, NY, USA.
✉email: amk2012@med.cornell.edu

Neural encoding models of vision that approximate brain responses to images have gained popularity in human studies with the wide-spread adoption of non-invasive functional MRI (fMRI) techniques[1] and recent advances in large-scale publicly available fMRI datasets of human responses to visual stimuli[2]. These neuroscientific resources have become available at a time of ubiquitous deep learning applications in every aspect of science and technology, but particularly in image analysis[3–8]. Recent work has revealed some agreement between image representations in biological and artificial neural networks (ANNs)[9]. This is somewhat unsurprising, as ANNs were originally inspired by the principles of how the feed-forward cortical network processes visual information[10,11]. Understanding how the human brain, unarguably the most efficient and adaptable learning system in the known universe, processes incoming information will no doubt lead to breakthroughs in neuroscience and artificial intelligence alike.

The functions and response properties of the visual cortex, with its central evolutionary role and ease of experimental perturbation, have been extensively studied[12–16]. Regions that respond to evolutionarily important content, like faces, bodies and places are relatively consistent across different individuals in their existence and spatial locations within the brain. Other regions, like those that respond to evolutionarily later content, like text/words, are more variable across individuals and are more experience dependent[17]. Recent work, including ours, has focused on investigating inter-individual differences in how brains process incoming stimuli[18]. One paper, in particular, revealed variations in neural and behavioral responses to auditory stimuli that are related to an individual's level of paranoia[19] while another showed that measuring brain responses to a video of naturalistic stimuli could amplify inter-individual variability in behaviorally relevant networks compared to task-free paradigms[20].

There are an increasing number of densely-sampled fMRI datasets, i.e. large amounts of data collected from a single individual, which enable both predicting brain response from natural images and, in turn, identifying natural images from brain activity patterns[2,21,22]. Accurate individual-level voxel-wise and region-wise encoding models can be created using thousands of training data provided by these datasets[23–26]. However, due to the excessive resources required to obtain large data from one individual, such experiments are usually restricted to less than 10 subjects and thus far cannot be used to predict a novel individual's responses. Population-level encoding models can be created by averaging densely-sampled individual encoding models or trained using pooled data from all subjects;[2,26] however, individual differences will be obscured using this approach. Compared to the number of publications that present encoding models built with large-scale fMRI data, work that utilizes small data to build encoding models is relatively limited. In one such example, Wen et al. (2018) used voxel-wise encoding models trained with 10 h of movie watching fMRI data as a prior to guide the estimation of the encoding model parameters for a novel individual for which they used a relatively small amount of movie watching data. There, they found that the encoding models trained in this way can achieve similar prediction accuracy as the 10 h trained models for that particular individual[27]. While useful, this work did not examine whether or not the pattern of inter-individual responses in the resulting encoding models was preserved. There is a clear need for a tool that can use previously collected densely-sampled data to efficiently create encoding models using small amounts of data in novel individuals, while preserving the uniqueness of that individual's brain responses to external stimuli.

In this work, we create and assess the accuracy of an ensemble encoding approach that uses existing individual encoding models pretrained with densely-sampled data to predict brain responses to visual stimuli in novel individuals. We quantify the specific number of image-response pairs that needs to be collected in the prospective individual to train a model and obtain accuracy similar to a densely-trained encoding model. Besides good accuracy, we also aim to make the model personalized, i.e. demonstrate the model's ability to preserve the inter-individual differences of measured responses. Most importantly for practical reasons, we quantified the accuracy of our ensemble encoding model when applied to novel individuals undergoing several domain shifts in the data and validated that this modeling approach could be used to efficiently and accurately create personalized encoding models. Finally, we demonstrated one potential application of the ensemble encoding models using a previously established NeuroGen framework to discover the inter-individual differences in several face area responses to animals and humans[18]. This shows the ensemble encoding models may be used to create images optimized to achieve maximal brain responses in a specific person in prospective experiments designed to explore inter-individual differences in visual processing.

## Results

The analysis was performed using two different datasets - the Natural Scenes Dataset (NSD) and the NeuroGen dataset (see "Materials and methods" for details). In short, the NSD dataset consists of ~24K pairs of images and corresponding brain responses from 8 individuals (6 female, age 19–32 years) who underwent 30–40 fMRIs while viewing natural scenes. The NeuroGen dataset consists of data from 6 individuals (5 female, age 19–25 years) who underwent two fMRI scans while viewing 800 images total. Both sets of data were used to train various encoding models that predict the region-level brain responses to an image, see "Materials and methods" for details. For the 8 NSD individuals, we created five different encoding models for each of four brain regions, including an early visual area - ventral V1 (V1v), and three late visual regions - fusiform face area 1 (FFA1), extrastriate body area (EBA) and parahippocampal place area (PPA), shown in Supplementary Fig. 1a. The five encoding models are (1) individual-20K model, which has the model architecture shown in Fig. 1a and is trained using all available data for a given individual, i.e. 20–24K image-brain response pairs, (2) scratch model, which shares the same architecture as the individual-20K model but is trained on only a subset of the available training data, i.e. 10 to 800 image-brain response pairs, (3) finetuned model, which is identical to the scratch model but the model weights of the linear readout were initialized using the average of the individual-20K model weights from the 7 other NSD individuals, (4) linear ensemble, which fits a linear model to predict the 8th NSD individual's measured responses from the 7 other individual-20K models' predictions, as shown in Figs. 1b and 5) average ensemble, which predicts the 8th NSD individual's responses as the average of the other 7 NSD individual-20K models' predictions. We consider the individual-20K models to be the gold standard reference model when assessing model performance for the NSD dataset. As there is no large-scale data available for the NeuroGen individuals, we created only the scratch, finetuned, linear ensemble and average ensemble encoding models. The scratch models for NeuroGen individuals were trained using only NeuroGen data, while the latter three were based on the 8 NSD individual-20K models and are thus considered to be out-of-distribution. Models were evaluated in two ways: (1) prediction accuracy calculated using Pearson correlation between the predicted activations and the measured activations from fMRI data and (2) prediction consistency calculated by the Pearson correlation between the inter-subject correlation (ISC) of

**a** Encoding model architecture

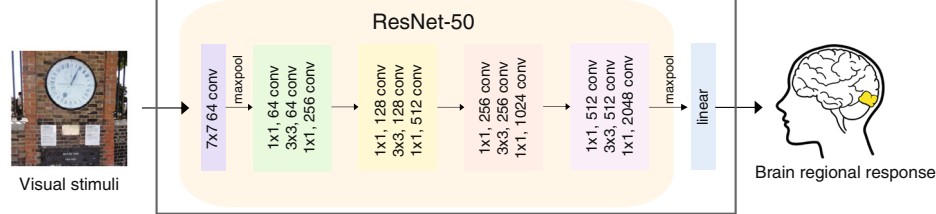

**b** Ensemble model architecture

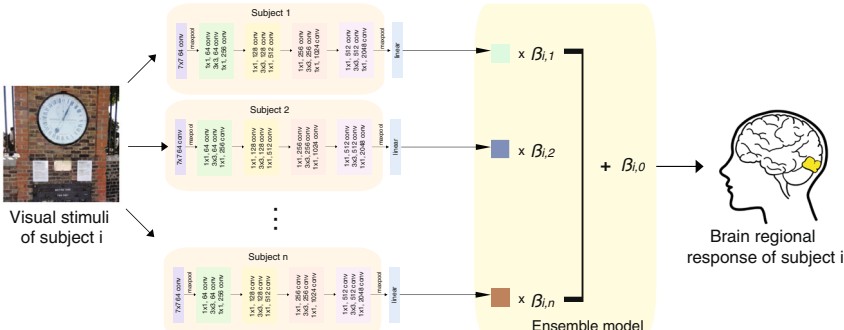

**Fig. 1 Encoding model and ensemble model architecture. a** The encoding model architecture. A feature extractor adopted from ResNet-50[8] extracts features from the input image and a linear readout maps the extracted features to the responses for a specific brain region. **b** The ensemble model architecture. A group of $n$ pretrained encoding models (where $n = 7$ or 8) are used to obtain a set of predicted activations, which are combined via a linear model fitted via ordinary least squares to optimally predict the query subject's regional brain response.

the predicted activations (i.e., the correlation between the predictions of each pair of subjects) and the ISC of measured activations. The latter measure quantifies the capability of the model to preserve inter-individual differences in brain responses. We note that while most encoding models in the literature use fixed, pre-trained feature extractors, we take an approach of fine-tuning the feature extractor (starting from the fixed, pre-trained weights) in order to achieve a more accurate encoding model (see Supplemental Fig. 4). This approach may change the interpretation of the model, but model inference was not the aim of this work—rather it was to provide a way to achieve the most accurate encoding models for a prospective individual using small amounts of data. An exhaustive comparison of the feature extractors of the encoding models used here to neural representations will be the object of future work.

**Validation of the individual-20K models**. Individual-20K model accuracies for regions FFA1, EBA, PPA and V1v for all 8 NSD subjects are shown in Supplementary Fig. 1b. Generally, we obtain good prediction accuracy, as measured via Pearson correlation between predicted and true brain activity in the hold-out set of test images, across all subjects for all regions. The mean accuracy for FFA1 is 0.531 with standard deviation (SD) 0.087; for EBA, the mean accuracy is 0.463 with SD 0.091; for PPA, the mean accuracy is 0.621 with SD 0.078; and for V1v, the mean accuracy is 0.608 and SD is 0.059. Supplementary Fig. 1c shows the top 10 images in the test set that have the highest predicted activation for each of the 4 regions for subject 1 (S1, first row) and subject 2 (S2, second row); subjects 3 through 8 are shown in Supplementary Fig. 2. We observe that the 10 images with highest predicted activity largely reflect the expected properties associated with activation in these regions. For example, almost all top images for FFA1 are human or animal faces; top images for EBA are people engaging in various sports; top images for PPA are all indoor scenes; and top images for V1v contain an abundance of texture and color. Although there are some common top images across subjects, there is also quite a bit of variability,

which supports the notion of individual differences in response patterns.

**Comparison of encoding model accuracies in the NSD dataset**. The scratch, finetuned, linear ensemble and average ensemble encoding models' accuracies across the 8 NSD individuals for varied training data sizes are provided in Fig. 2. Each boxplot illustrates the distribution of accuracy values over the 8 individuals in the NSD dataset. For comparison, the individual-20K accuracy is provided via the purple boxplot, after the break in the $x$-axis required to indicate the large size of the individuals' full set of image-response pairs. As the average ensemble model doesn't require any training data from the individual in question, its accuracy is indicated via the orange boxplot on the right. Unsurprisingly, the accuracies for the scratch, finetuned, and linear ensemble models increase with training data size.

There is an obvious accuracy gap between finetuned models and scratch models, indicating that model performance benefits from the initialization at the group average readout. Interestingly, both the linear ensemble and average ensemble models consistently outperform the scratch and finetuned models. In fact, when there are more than 100 image-response pairs available for training, there are no significant differences between the linear ensemble models and individual-20K models for any region (Wilcoxon tests with $p > 0.05$).

To further explore the differences in each of the models, we performed a detailed comparison between the encoding models trained on a dataset of 300 image-response pairs, across-subject noise ceiling (NC) and within-subject NC, see Fig. 3a–d. The across-subject NC is calculated as the Pearson correlation between subject's measured responses and the across-subject average of the other subjects' measured responses in the test set, while the within-subject NC is calculated as the Pearson correlation between the average of two measured responses for a subject and their third measured response in the test set. A training size of 300 was chosen because it is reasonable amount of

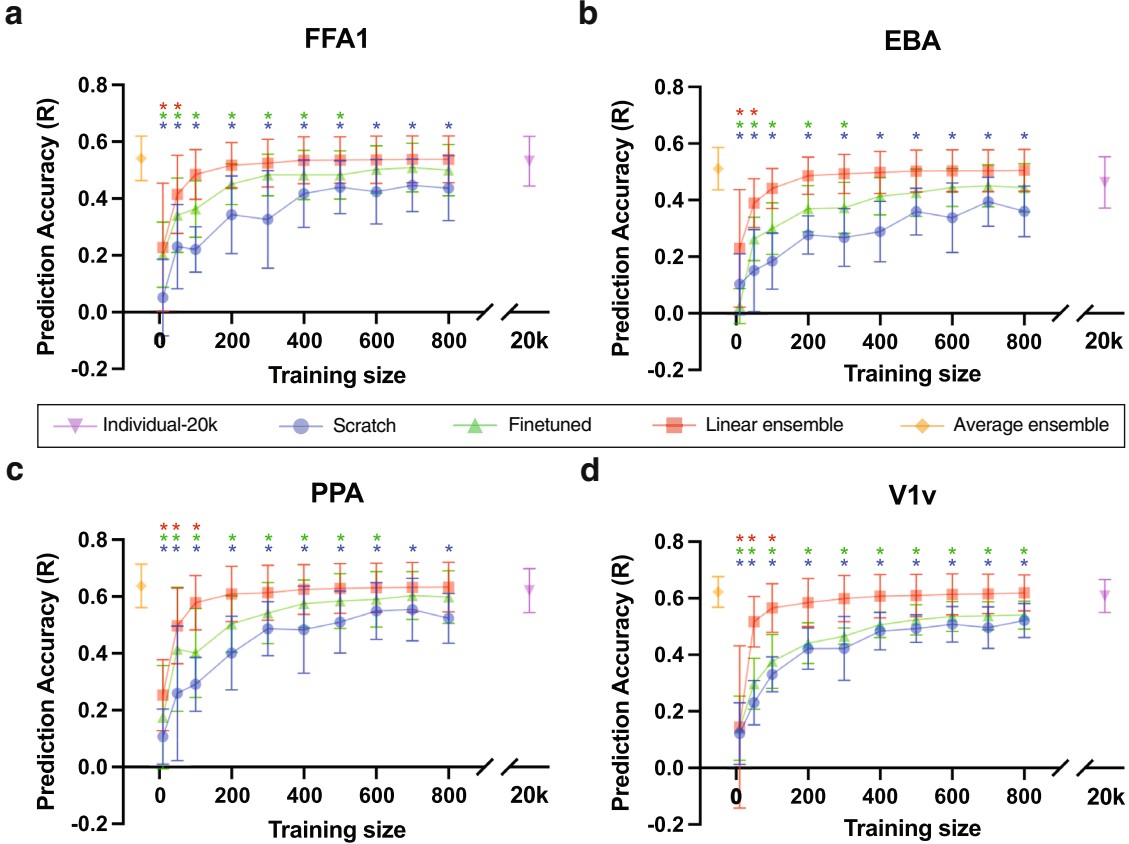

**Fig. 2 Encoding model accuracies for FFA1, EBA, PPA and V1v regions, by the size of the training dataset (x-axis). a–d** Boxplots indicate encoding model test accuracy across 8 NSD subjects for the scratch (blue), finetuned (green) and linear ensemble (red) models as training size increases from 10 to 800 image-response pairs. Individual-20K model accuracy is illustrated via the purple boxplot (training size of 20K) and the average ensemble model accuracy is illustrated via the orange boxplot (training size of 0). Asterisks, colored according to the model in question, indicate a significant difference (Wilcoxon tests, $p < 0.05$) in those models from the gold-standard individual-20K models.

images that one participant can view during a 60 min MRI scan and the linear ensemble and scratch model accuracies largely plateau around a training size of 300. There were no significant differences in accuracy between the individual-20K and ensemble models (Friedman's test with FDR correction $p > 0.05$). The scratch models trained using 300 image-response pairs always had significantly lower accuracy for all regions compared to individual-20K models, and in most cases also had significantly lower accuracy than ensemble models. The finetuned models performed better than scratch models but were still significantly lower than average ensemble models for all regions. The across-subject NC, higher than the within-individual NC due to a better signal-to-noise ratio (SNR) of the measurements, behaved as a upper bound for the prediction accuracy and was always significantly higher than scratch and finetuned models. The individual-20K model only showed significantly lower accuracies than the across-subject NC upper bound for one region (EBA); similarly, the ensemble models also showed significantly lower accuracies compared to this upper bound only in PPA. The within-subject NC was always significantly lower than the individual-20K and the ensemble model accuracies. We also compared linear ensemble models with deep-ensembled individual-20K models and found similar performances (see Supplementary Fig. 8). These results demonstrate the ability of the linear ensemble model (trained on only a few hundred image-response pairs from the novel individual) and average ensemble model (trained on no data from the novel individual) perform as well as

models trained on very large data ($\sim$20K) in predicting the regional brain responses of the novel individual to a given image.

**Comparison of encoding model accuracies in the out-of-distribution NeuroGen dataset**. We assessed the performance of the scratch, finetuned, linear ensemble, average ensemble encoding models, and the across- and within-subject NCs using the NeuroGen dataset. The within-subject NC for NeuroGen subjects is calculated as the Pearson correlation between two measured responses in the test set. Importantly, the two NC metrics were not significantly different from the encoding model accuracies, indicating that overall the encoding model accuracy is relatively good. This experiment introduces several domain shifts, including the modeling of different individuals (not NSD individuals), (some) different visual stimuli, different MRI scanner/strength (NSD having 7T vs NeuroGen having 3T) and fMRI parameters (TR, voxel size etc). Thus, we did anticipate a drop in the NeuroGen individuals' prediction accuracies compared to the within-distribution NSD individuals' accuracies for the finetuned, linear and average ensemble models. Despite experiencing some of this anticipated drop, overall the prediction accuracies do remain at a good level, see Fig. 3e–h. For EBA and PPA, both ensemble models both had significantly higher accuracy than scratch models; for PPA the ensemble models also outperformed the finetuned models (Friedman's test with FDR correction $p > 0.05$). No significant differences were found between the linear

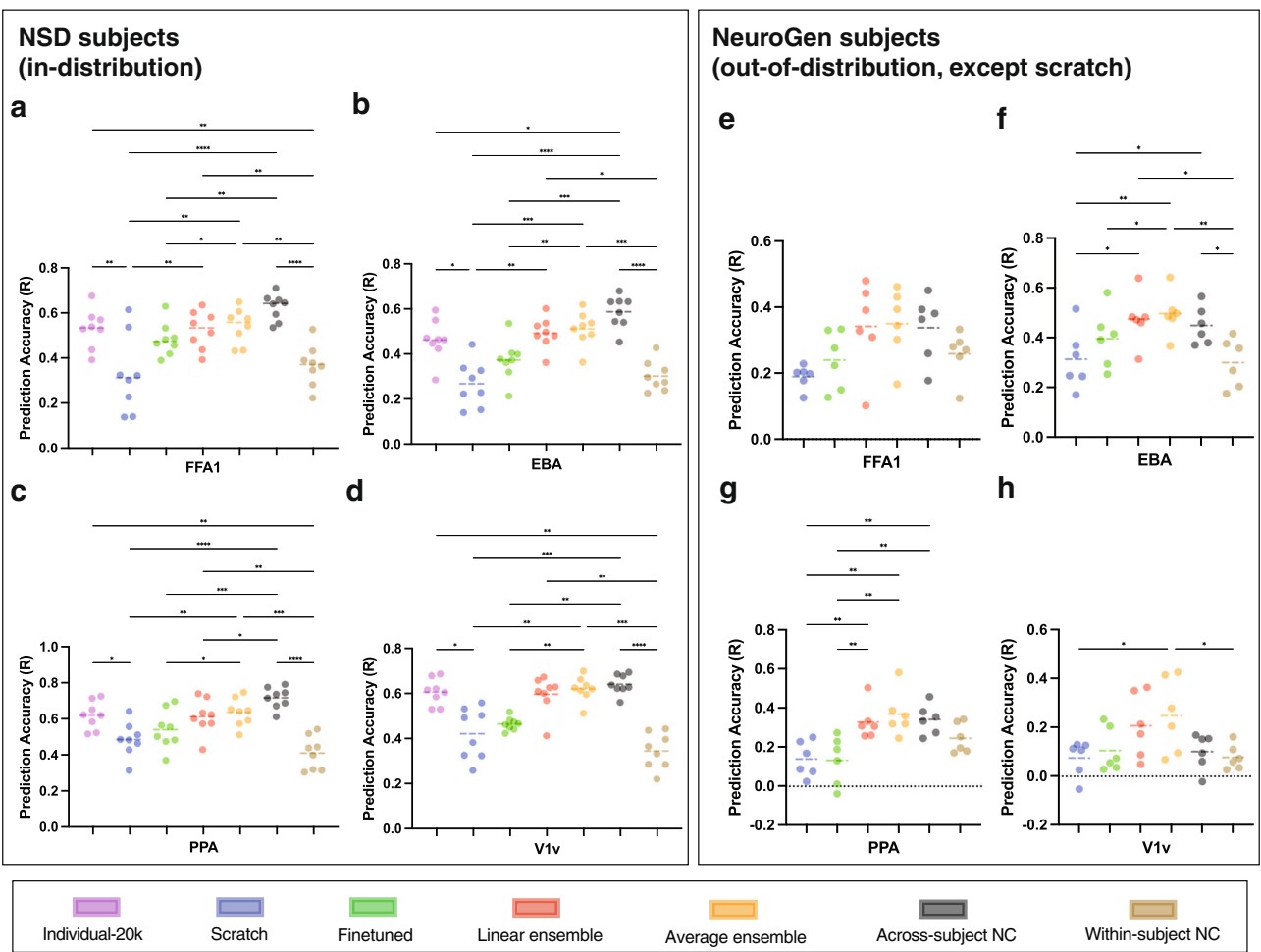

**Fig. 3 Encoding model prediction accuracies.** Accuracies for the 8 NSD individuals **a** FFA1, **b** EBA, **c** PPA, **d** V1v, and 6 NeuroGen subjects **e** FFA1, **f** EBA, **g** PPA, **h** V1v, measured via Pearson's correlation between predicted and observed regional activity across a set of test images for 4 regions - FFA1, EBA, PPA and V1v. Individual-20K models are illustrated in purple, scratch models in blue, finetuned in green, linear ensemble in orange and average ensemble in yellow. Scratch, finetuned and linear ensemble models were trained with a dataset of 300 image-response pairs from the novel subject in question. The across-subject noise ceiling (NC), representing the correlation between the average measured test-set responses of the other subjects with the measured test-set responses of the subject in question, and within-subject NC, representing the correlation of the individual's measured image responses, are shown in black and brown, respectively. Black bars at the top of the panels indicate significant differences in accuracy across the indicated pairs of models (Friedman's test with FDR corrected $p < 0.05$). The correspondence between the number of asterisks and the $p$-value: * - $p \leq 0.05$, ** - $p \leq 0.01$, *** - $p \leq 0.001$, **** - $p \leq 0.0001$.

and average ensemble models' performances for any region. The ensemble models did not have significantly lower accuracy compared to either NC for any model (and in fact had significantly higher accuracy compared to the within-subject NC for EBA), while scratch and finetuned models had significantly lower accuracies than the across-subject NC in PPA. In addition, since NeuroGen dataset contains both natural and synthetic images while the individual-20K models were trained only on natural images, we also examined the prediction accuracy of the natural and synthetic images separately, and there were no different from each other (Welch's t test, two-tailed $p = 0.9599$). Finally, we verified that the features of the images with the highest predicted activity from the NeuroGen's encoding models agreed with expectations, e.g. top FFA1 images were faces, see Supplementary Fig. 3.

**Preservation of inter-individual differences of brain responses within the encoding models.** An encoding model should not only be accurate, but it should also preserve inter-individual

differences in response patterns as much as possible. To illustrate our models' abilities to preserve inter-individual differences, we constructed and calculated the *prediction consistency*, which is the correlation of the ISC of measured activations (ISC-measurement) and the ISC of predicted activations (ISC-prediction) over the images in the test set. This metric of prediction consistency quantifies how well the encoding model predictions preserve the measured between-subjects differences in brain responses. The average ensemble model does not preserve inter-individual differences in predictions and thus can serve as a lower-bound for comparing model consistency. Since the ISCs of the average ensemble model's predicted activities will all be 1 or near 1 (with leave-one-out training), the prediction consistency of the average ensemble models is ~0 or undefined. The ISC-prediction and ISC-measurement were calculated for each pair of subjects within the two datasets (NSD and NeuroGen), for each of the four brain regions of interest. We first demonstrated that there are reliable inter-individual differences in brain responses, evidenced by the variability of the top images for NSD subjects shown in Supplementary Fig. 2, and the low ISC-measurement values in Fig. 4

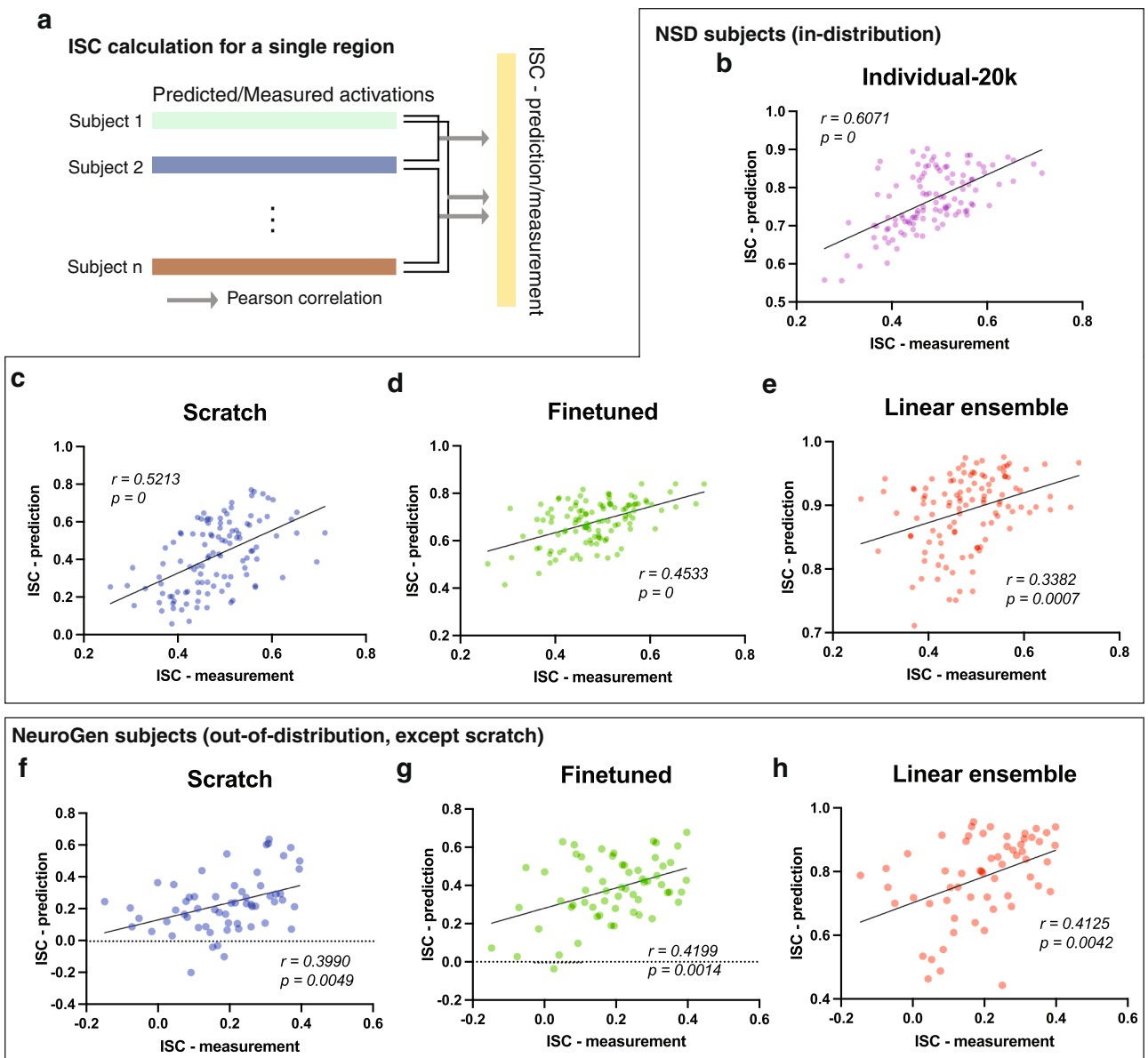

**Fig. 4 Preservation of inter-individual differences within the encoding models. a** Inter-subject correlations (ISC) are computed as the Pearson correlation between every pair of subjects' predicted (ISC-prediction) or measured (ISC-measurement) brain responses, calculated for each of four brain regions separately. The scatter plots show prediction consistency, which is the relationship between the ISC-measurement and ISC-prediction values across every pair of subjects for the images in the test set; the same region's measured/predicted activity is used for both subjects. The number of points in each scatter plot are therefore the number of brain regions (4) * number of pairs of individuals (for NSD this is $(7 \times 8)/2 = 23$ and for NeuroGen this is $(6 \times 5)/2 = 15$). The scratch, finetuned, and linear ensemble models here were created using a training dataset of 300 image-response pairs for NSD subjects. **b-e** represent the NSD subjects while **f-h** represent the NeuroGen subjects. Individual-20K models are not available for the NeuroGen individuals. The predicted responses have better SNR than the measured responses (due to the noise in the fMRI responses that is smoothed in the encoding models), thus note the x and y-axis ranges are quite different. All p values (calculated via permutation testing) are one-tailed and Bonferroni corrected for multiple comparisons.

which were shown to be pretty robust in Supplementary Fig. 9. Unsurprisingly, we observed that the NSD dataset's individual-20K model has high prediction consistency, as this model's predictions best preserve the observed inter-individual differences in brain responses (see Fig. 4b, Pearson's $r = 0.6071$, $p = 0$). Despite the overall lower accuracies of the scratch and finetuned models, they do preserve a good amount of inter-individual differences (Fig. 4c, Pearson's $r = 0.5213$, $p = 0$, and Fig. 4d, Pearson's $r = 0.4533$, $p = 0$). The linear ensemble model also has good preservation of inter-individual differences (Pearson's $r = 0.3382$, $p = 0.0007$), see Fig. 4e. When evaluating the NeuroGen dataset,

all three models have similar prediction consistencies (Pearson's $r = 0.399$–$0.4199$, $p = 0.0014$–$0.0049$). These results demonstrate that, out of the four models using small data, the linear ensemble model has the best balance of accuracy and preservation of inter-individual differences. Importantly, it also achieves accuracy similar to the individual-20K model using a training dataset that is only 1.5% the size of the larger model's training data.

We assessed the reliability of the prediction consistency and ISC-measurement metrics by randomly subsampling the data 1000 times and recalculating the ISC-measurement and the individual-20K model's prediction consistency. We indeed see

robustness of the ISC-measurement, with the relative errors being mostly within 3% of their original values, as well as the prediction consistency which varied within only 10% of the original value (see Supplementary Fig. 9). We also investigated how varying the number of individuals used in the linear ensemble model affected prediction accuracy and consistency, see Supplementary Fig. 5. We see that there is an increase in accuracy from using one to two individuals' models in the linear ensemble for the NSD subjects, but additional individuals beyond two do not further increase accuracy. For NeuroGen, linear ensemble model accuracy does not seem to increase with increasing number of individuals' models. However, for both datasets, the prediction consistency, or the preservation of inter-individual differences in the predictions, increases with the number of individuals' models used in the linear ensemble. This indicates the importance of using at least several individuals in a linear ensemble in order to maintain inter-individual differences in a linear ensemble approach. Supplementary Fig. 6 further visualizes the linear ensemble model coefficients across all subjects and brain regions; we can see that there is wide variability in the weights for the linear ensemble models over the individual and brain region in question.

**Application: linear ensemble models capture individual regional preferences when integrated into the NeuroGen algorithm**. We used our previously established NeuroGen framework to demonstrate an application of the proposed linear ensemble encoding model[18]. NeuroGen, originally composed of an individual-20K encoding model and a image generator (see Fig. 5a), synthesizes images that achieve a desired target activity in a specific region or regions of interest. It was previously used to uncover inter-individual and inter-regional differences in animal/dog face vs human face preferences, which was validated with measured preferences in the fMRI data[18]. We set out to test if the NeuroGen framework can be equally as useful using a linear ensemble encoding model trained with small data. To that end, within NeuroGen we replaced the individual-20K model with the linear ensemble model and produced the top 10 synthetic images that maximize activation for each of three face regions: OFA, FFA1 and FFA2, see Fig. 5b–d (more examples are provided in Supplementary Fig. 7). First, we qualitatively observed that the synthetic images contain animal or dog faces as expected. Note: each of the top 10 images uses a different one-hot encoded class vector but some subjects/regions top 10 image sets may have the same class vector leading to similar-looking images across subjects/regions top 10 image sets (for example, the person wearing a red wig). Additionally, BigGAN has a truncation parameter that allows generation of sets of images with varying balance of variety and fidelity. If the truncation parameter is large, the images will have more variety but look less realistic than if the truncation parameter is small. Our images were generated using truncation parameter 0.1 in order to enforce more realistic images, of course at the cost of decreased variability within a category. Next, we correlated (1) the t-statistic of regional observed activity (from the fMRI data) in response to images of animal faces vs human faces and (2) the animal face vs human face ratio in the top 10 synthetic images, see Fig. 5e. We observed a significant positive correlation between the two values (Pearson's $r = 0.5443$, two-tailed $p = 0.0195$), indicating that the NeuroGen framework using the linear ensemble encoding model still preserves inter-individual and inter-regional measured differences in animal vs human preference. We tested the reliability of this result by randomly sub-sampling 1000 splits of the NeuroGen data and re-calculating the animal-person t-statistic and its correlation with the top 10 synthetic images. We found it to be relatively robust

with a coefficient of variation of ±12.3%. The observed correlations are also similar when using more synthetic images, e.g. top 100, to calculate the synthetic image ratio, again strengthening the point that individual/regional differences can be captured even in the top 10 synthetic images. Of course, when using the average ensemble encoding model there are no meaningful results as the top 10 synthetic images (and ratios) are identical for every NeuroGen subject. These results demonstrate the relatively robust nature of the NeuroGen framework for discovering inter-individual differences in neural representations of visual stimuli in novel individuals. Validating the NeuroGen framework using an encoding model trained with small data from a novel individual demonstrates how it may be leveraged for prospective experimental design.

## Discussion

Here we propose a visual encoding model framework that linearly combines outputs from several individuals' existing encoding models (trained on densely-sampled NSD data) to predict a novel individual's brain responses to a given image. We show that the proposed linear ensemble model, trained using a relatively small number of image-response pairs for the novel individual ($\sim 300$, roughly equivalent to 40 min of fMRI), achieves accuracy similar to encoding models trained on a very large number of image-response pairs from that individual ($\sim 20K$, roughly 35–40 h of fMRI). Importantly, the linear ensemble model predictions also preserve a pattern of inter-individual differences in measured responses that was on par with what is observed using the encoding models trained on very large data. We also validated the accuracy of the linear ensemble model on prospectively collected, out-of-sample data; despite several domain shifts we still obtained good accuracy. Using the linear ensemble encoding models within NeuroGen, a synthetic image generator previously proposed as a tool for discovery neuroscience, we reproduced the measured individual/regional differences in animal face vs human face preference for three face regions. These results suggest that the linear ensemble model can be used to efficiently create accurate, personalized encoding models able to be used within our NeuroGen framework to optimize synthetic images for prospective human vision experiments.

Neural encoding and decoding models have long been used to characterize and predict how sensory, cognitive or motor information is spatially represented in the brain[1,28,29]. Recent work has revealed that understanding the inter-individual differences in responses to naturalistic stimuli may shed light on behavioral or pathological variability in humans[19,20,30] and monkeys[31]. Having high-quality and large-scale stimuli-response data is critical to building accurate and useful encoding or decoding models, but due to the massive cost in time and resources there are only a few such datasets available[2,22]. Retrospective analyses of these datasets are constrained by the original parameters of the experiment and the characteristics of the stimuli presented within them. If it is not possible to test a specific hypothesis with the existing data, scientists will need to collect new data on novel individuals. Our approach proposed here aims to bootstrap existing large-scale datasets to improve the starting point of these prospective experiments by providing a more accurate baseline visual encoding model that also preserves inter-individual differences in response patterns. Furthermore, we provide quantitatively derived guidelines for how many images are needed to achieve accuracy similar to encoding models trained on very large-scale data in Fig. 2, and the effects of varying the number of individuals' pretrained models used in the linear ensemble model in Supplementary Fig. 5. For the latter analysis, we conjecture that, since we are using the higher SNR individual-20K encoding

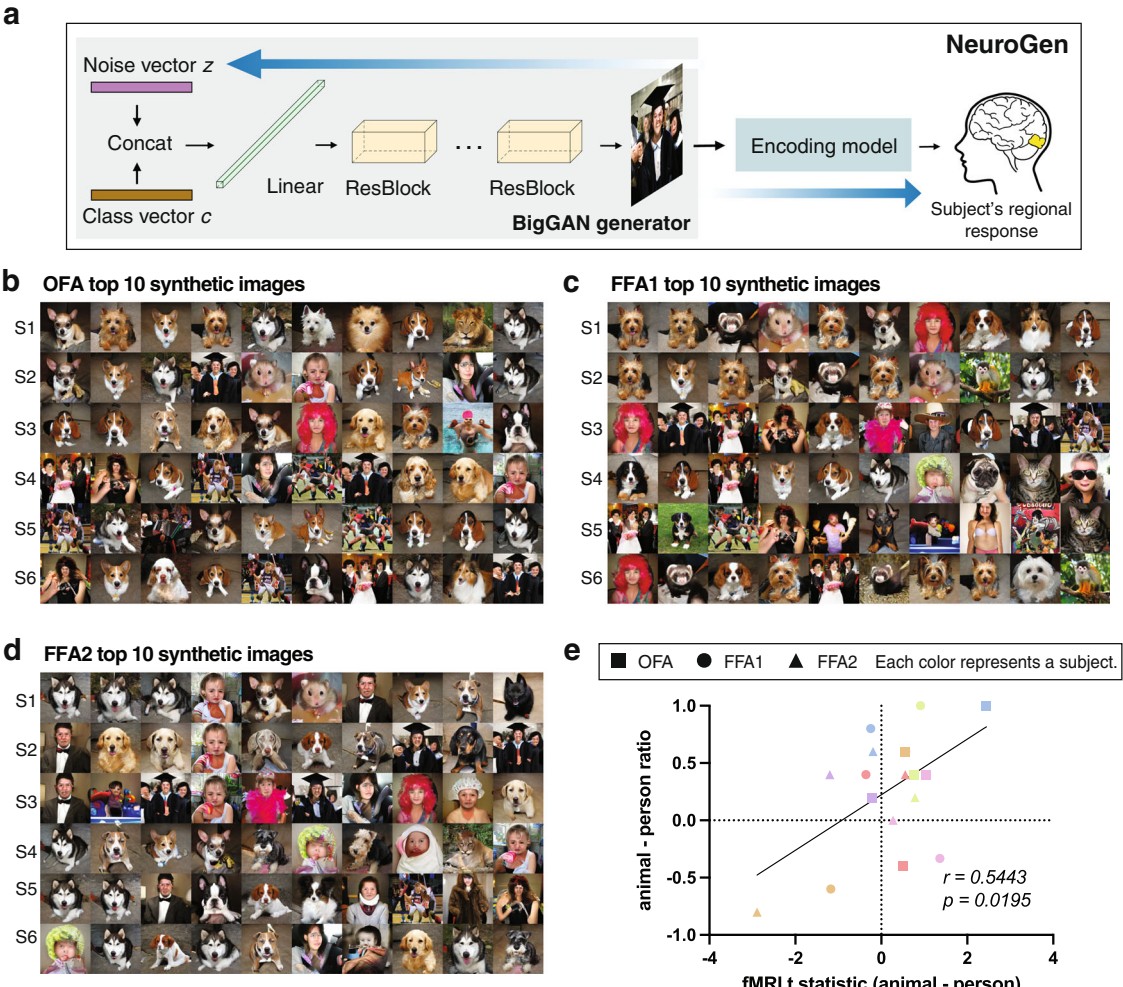

**Fig. 5 Linear ensemble models capture individual regional preferences in NeuroGen framework. a** The NeuroGen framework concatenates a image generator (BigGAN) with an encoding model to synthesize images that achieve a desired predicted response in one or more brain regions. One synthetic image is generated from a 1000 dimension, one-hot encoded class vector (corresponding to one class in ImageNet) and a noise vector, which will be identified through optimization Different synthetic images may have different class vectors. The output image is then fed into the encoding model to obtain a region's predicted response for that image. **b–d** Synthetic images from the version of NeuroGen using the linear ensemble encoding model; the 10 images were designed to maximize activation for each of the six NeuroGen individuals (one individual per row) in OFA, FFA1 and FFA2 regions, respectively. **e** A scatter plot indicating for each of the 3 face regions for each of the six NeuroGen individuals, the observed t-statistic of OFA, FFA1, and FFA2's responses to images of animal versus human faces (x-axis) and the ratio of the number of animal images minus the number of human images, divided by the sum of these two numbers, out of their top 10 synthetic images created via NeuroGen (y-axis). Each color represents a subject and each shape represents a different face region.

model predictions in the linear ensemble (and not the lower SNR measured fMRI responses), that even using one individual's encoding model gives us the highest prediction accuracy we can get (without the prediction consistency). Thus, adding more individuals' encoding models to the linear ensemble does not increase model accuracy but does increase prediction consistency, i.e. better preserves the inter-individual relationship of responses across individuals. Relatedly, we found that the linear/average ensemble models perform as well as the individual-20k models for the NSD data, which could either be explained by the individual-20K overfitting (although not likely from our assessments of model fit) or the fact that all three models are approaching the ceiling of possible accuracy determined by the relatively low SNR of the fMRI data.

Linear models, with their simplicity and desirable statistical properties, have been a fundamental statistical/machine learning approach[32] with many applications in network, cognitive, visual neuroscience[23,33,34]. The gold-standard validation of statistical or

machine learning models is demonstrating their accuracy is robust to domain or distribution shift of the underlying data[35]. Here, our domain shift happens in three major ways—first is the data acquisition (scanner, scanner strength, repetition time, voxel size, etc.), second is the individuals undergoing the experiments were not identical and the third was the type of images shown to the individuals during the scan (natural versus natural and synthetic). Despite these several major shifts in data characteristics, the linear ensemble models are still able to obtain a good level of accuracy that outperforms scratch trained or finetuned models. The drop in accuracy from the NSD dataset was not identical for each region, some regions, e.g. EBA, appeared more robust to domain shift than others. Future work could explore using nonlinear approaches to ensemble modeling that may allow better preservation of accuracy across varied experimental conditions.

Prediction accuracy is the most common metric when evaluating neural encoding models. However, few works have investigated how the models preserve inter-individual differences. For example,

a model only predicting the population average (or just using responses from one other individual) may have high prediction accuracy, however, these models do not preserve individuality of a subject's brain responses. This work focused on evaluating both accuracy and individuality (here called consistency) with the aim to create models that are both accurate and personalized. We generally found a trade-off between prediction accuracy and consistency, which may explain why the linear ensemble models (which maintain consistency) did not outperform the average ensemble models (which do not maintain consistency). Across individuals, consistency and model accuracy are somewhat entwined. If a query subject's responses are very similar to the reference individuals' responses (and thus have high ISC-measurement and high ISC-prediction), the accuracy of the ensemble model will also be high. On the other hand, if a query subject's responses are uncorrelated with the reference individuals, the ISC-measurement and accuracy will be low but the ISC-prediction will remain high (as it will be based on the reference individual's models that are all very similar to one another). This is one consideration to make when interpreting the consistency metric. However, they are clearly not entirely overlapping, as evidenced by two facts: that the average ensemble has similar accuracy compared to the linear ensemble but undefined consistency and Supplementary Fig. 5, which demonstrates that maximal linear ensemble model accuracy is achieved with only 1 or 2 reference subjects while model consistency continues to increase as more and more individuals are added to the model.

Our previously developed NeuroGen framework[18] was used to generate synthetic images that were predicted by a given encoding model to achieve a targeted pattern of activity in a specific brain region, e.g., maximizing the activity in the FFA1. Unlike similar frameworks developed for monkey or mouse models[36-38], it is difficult to directly optimize stimuli for humans in real time while they are undergoing fMRI. An achievable alternative may be to fine-tune our linear ensemble model for a prospective individual using image-response data collected at the first scan, which could then be inserted into the NeuroGen framework to create personalized, optimized synthetic images shown to the same individual at a second scan. In this scenario, if the linear ensemble encoding model accuracy drops for synthetic images then the NeuroGen framework will fail. In our current results, we didn't find a significant difference between the linear ensemble models' prediction accuracy for natural and synthetic images, which provides essential evidence that the linear ensemble model accuracy is not different between the natural and synthetic images and is thus an essential part of prospective experiments utilizing NeuroGen.

Interestingly, we demonstrated that the inter-individual preference in face regions for animal versus human faces was successfully replicated with the NeuroGen framework utilizing the personalised linear-ensemble models for the prospective NeuroGen individuals, despite the fact that the linear ensemble models had somewhat lower accuracy compared to the NSD individuals. This demonstrates the robustness of the NeuroGen framework to drops in the accuracy of the encoding model used within, and suggests that our previously identified inter-individual differences in face area responses to images containing animal versus human faces exists in novel individuals and can be quantified using the top 10 images created by NeuroGen.

This work proposes and validates an ensemble framework that uses previously collected, densely-sampled data to efficiently create accurate, personalized encoding models and, subsequently, optimized synthetic images for novel individuals via NeuroGen. Importantly, we validate that the encoding models can be applied in novel experimental conditions and that they did not have different accuracy for predicting responses to natural versus synthetic images. Future work will use this framework to prospectively investigate inter-individual differences in visual encoding and create personalized synthetic images designed to achieve a targeted pattern of brain activity within a specific individual.

## Materials and methods
### Data description
*Natural Scenes Dataset.* The individual encoding models were created using the Natural Scenes Dataset (NSD)[2]. The informed consent for all subjects was obtained by NSD. Our data usage was approved by NSD, and complies with all relevant ethical regulations for work with human participants. NSD contains densely-sampled fMRI data from eight participants (6 female, age 19–32 years). Each subject viewed 9000–10,000 distinct color natural scenes with 2–3 repeats per scene over the course of 30–40 7T MRI sessions (whole-brain gradient-echo EPI, 1.8 mm iso-voxel and 1.6 s TR). The images that subjects viewed (3 s on and 1 s off) were from the Microsoft Common Objects in Context (COCO) database[39] with a square crop resized to 8.4° × 8.4°. A set of 1000 images were shared across all subjects, while the remaining images for each individual were mutually exclusive. Subjects were asked to fixate centrally and perform a long-term continuous image recognition task (inf-back) to encourage maintenance of attention.

The fMRI data were pre-processed to correct for slice time differences and head motion using temporal interpolation and spatial interpolation. Then the single-trial beta weights representing the voxel-wise response to the image presented was estimated using a general linear model (GLM). There are three steps for the GLM: the first is to estimate the voxel-specific hemodynamic response functions (HRFs); the second is to apply the GLMdenoise technique[40,41] to the single-trial GLM framework; and the third is to use an efficient ridge regression[42] to regularize and improve the accuracy of the beta weights, which represent activation in response to the image. FreeSurfer was used to reconstruct the cortical surface, and both volume- and surface-based versions of the voxel-wise response maps were created. The functional localizer (fLoc) data was used to create contrast maps (voxel-wise t-statistics) of responses to specific object categories, and region boundaries were then manually drawn on inflated surface maps by identifying contiguous regions of high contrast in the expected cortical location, and thresholding to include all vertices with contrast > 0 within that boundary. Early visual ROIs were defined manually using retinotopic mapping data on the cortical surface. Surface-defined regions were projected back to fill in voxels within the gray matter ribbon. Region-wise image responses were then calculated by averaging the voxel-wise beta response maps over all voxels within a given region.

*NeuroGen Dataset.* To further prove the proposed encoding models are robust against domain shift and translatable to novel individuals, we collected prospective data we are calling the NeuroGen dataset. The study protocol is approved by an ethical standards committee on human experimentation, and written informed consent was obtained from all participants. The NeuroGen dataset contains data from two MRI sessions about 4 months apart. During the first session, six individuals (5 female, age 19–25 years) underwent MRI, including an anatomical T1 scan (0.9 mm iso-voxel), a functional category localizer to identify higher-order visual region boundaries (as in the NSD acquisition), and, finally, fMRI while viewing a set of 480 images. Two-hundred and forty of the images were selected from NSD image training set, and 240 were synthetic images created by NeuroGen[18], a generative framework that can create synthetic images designed to achieve a specific desired brain activity pattern (see Fig. 5a). During the second MRI session, the same six individuals underwent fMRI while viewing 336 images (half natural and half synthetic). In the second session, the images varied across individuals. The image viewing fMRI acquisition setup was replicated as closely as possible to the NSD acquisition, i.e. the images were square cropped and resized to 8.4° × 8.4° and were presented for 3 s on and 1 s off. Data were acquired on a GE MR750 3T scanner. The fMRI scans had posterior oblique-axial slices oriented to capture early visual areas and the ventral visual stream (gradient-echo EPI, 2.25 × 2.25 × 3.00 mm, 27 interleaved slices, TR = 1.45 s, TE = 32 ms, session-encoding in the A»P direction). EPI susceptibility distortion was estimated using pairs of spin-echo scans with reversed session-encoding directions[43]. Preprocessing included slice-timing correction with upsampling to 1 s TR, followed by a single-step spatial interpolation combining motion, distortion, and resampling to 2 mm isotropic voxels. GLMs were fitted identically as in the NSD description above. Retinotopic regions were defined on NSD subjects, from which a probabilistic map was created in surface-aligned fsaverage space. Probability maps for each ROI were then binarized (>0) and projected to each NeuroGen subject's surface and then functional voxel space.

### Encoding model architecture, building and quality assessment
*Model architectures.* Figure 1a illustrates the architecture of the individual-20K, scratch and finetuned encoding models, which takes an input image and predicts a particular visual region's average response (the mean response over the voxels in that region). The encoding model contains a feature extractor taken from ResNet-50[8] which extracts both low and high level image features via convolutional blocks, a global max-pooling layer and a linear readout layer which maps the features to brain response. The image feature maps (shape 7 × 7 × 2048) are derived from the final layer of the ResNet50 backbone. Since the weights of the feature extractor

derived from pretraining on ImageNet were not fixed but instead finetuned using the neural data, empirically we found that encoding models based on feature maps derived from later layers had better accuracy (compared to earlier layers) for both early visual areas and higher order regions. The model was built in PyTorch. We initialized the feature extractor with ImageNet[44] pretrained weights for the individual-20K, scratch and finetuned models. The linear readout was randomly initialized for the individual-20K and the scratch models, while for the 8th NSD subject's finetuned model it was initialized with the average weights from the individual-20K models across the remaining 7 NSD subjects and initialized with the average across all 8 NSD subjects for each NeuroGen subject. During training, the parameters in both the feature extractor and readout were updated by minimizing the mean square error between the predicted responses and the measured responses. The optimizer was AdamW and batch size was set to 32. Models were trained until the correlation between the predicted responses and measured responses in the validation set stopped increasing and demonstrated convergence.

*Linear and average ensemble models.* The framework for the linear ensemble model is shown in Fig. 1b. This model predicts an individual's response to image $S$ using a linear model applied to the predictions from the other subjects' pretrained individual-20K encoding models:

$$LE_i(S) = \beta_{i,0} + \sum_{j \in N_i} \beta_{i,j} \hat{r}_j(S) \tag{1}$$

where $\beta_{i,0}$ is the intercept for individual $i$'s model, $N_i$ indicates the set of indices not including individual $i$, $\beta_{i,j}$ is the coefficient for the $j$th individual-large encoding model in predicting individual $i$'s responses, and $\hat{r}_j(S)$ is the predicted activation in response to image $S$ for subject $j$'s individual-large encoding model.

For each NSD subject, the linear ensemble model was constructed using a leave-one-out method where the other 7 NSD subjects' pretrained individual-20K models were used. For the NeuroGen subjects, all 8 NSD subjects' pretrained individual-20K models were used. The linear ensemble weights were fitted via ordinary least squares using randomly selected samples (number equals to train size) from the original 20K training set. The training size for NeuroGen models is on average 560. The average ensemble model shares the same framework as linear ensemble but there is no model fitting involved—it merely averages the predicted responses from the individual-20K models from the other 7 NSD individuals (in the case the individual is from the NSD dataset) or all 8 NSD individuals (in the case the individual is from the NeuroGen dataset).

*Model building: Training set.* The training set for individual-20K models contains 8500 unique images (each shown 0-3x), corresponding to around 20,000 image-response pairs for each subject. The training sets for the NSD encoding models using small data, including the scratch, finetuned and linear ensemble models, were random subsets of the complete set of images. For NeuroGen subjects, the training set contains around 560 image-response pairs from the second MRI session. No brain-responses to the same image were averaged—each datapoint in the training set represented a single brain-response to a given image.

*Model building: validation set.* The validation set was identical for the same subject across all model types. For each NSD subject, we selected 500 images from their set of 9000 unique images that had at least two fMRI measurements. We averaged brain response maps for images that were shown to the subject twice; if an image was shown three times two of the corresponding brain responses were randomly selected and averaged. This ensured the SNR properties of the brain response maps would be consistent across images within a subject and across subjected within the NSD dataset. For the NeuroGen subjects, we selected 86 images from the second MRI session and obtained the average brain response of two image presentations, as in the NSD dataset.

*Model building: test set.* The test set was identical for the same subject across all model types. For each NSD subject, we selected the 766 images from the shared 1000 set of images that had at least 2 presentations per subject, over all subjects. We then averaged the brain responses from two different viewings of that image. For the NeuroGen subjects, the test set of brain responses corresponded to the 127 images from the first MRI session that were identical across all NeuroGen individuals and shown at least twice; the brain-response maps were again averaged over the two presentations of the image.

*Encoding model assessments.* Models were assessed in two ways—prediction accuracy and prediction consistency, i.e. preservation of inter-individual differences in brain responses. The model's prediction accuracy was calculated as the Pearson's correlation between the predicted and the measured responses across the test set of images. Wilcoxon tests were used to compare the prediction accuracy of models with different training data sizes to the "gold-standard" individual-20K models in Fig. 2. Friedman tests with FDR corrections for multiple comparisons were used to assess significant differences in prediction accuracy between different models in Fig. 3. A model's ability to preserve individual differences in image response patterns was assessed via prediction consistency, which is calculated as the Pearson correlation of the ISC of the predicted responses, or correlation of the predicted responses for every pair of individuals, and the ISC of the measured

responses within the test set of images. The second level $p$-values were calculated using 10,000 random permutation tests.

*Noise ceiling estimates.* We calculated two types of noise ceilings (NC) with which to compare our encoding models' prediction accuracies: across-subject NC and within-subject NC. Across-subject NC is computed as the Pearson correlation between subject's measured responses and the average of the other subjects' measured responses across the test data set. Within-subject NC calculations are slightly different across NSD and NeuroGen subjects. For NSD subjects, the within-subject NC is calculated as the Pearson correlation between the average of two measured responses and the third measured response. For NeuroGen subjects, the within-subject NC is calculated as the Pearson correlation between two measured responses in the test set as there are very few images that have three measured responses.

**NeuroGen: activation optimized image synthesis.** The NeuroGen framework, illustrated in Fig. 5a, concatenates an image generator (BigGAN-deep[45]) with an encoding model to generate synthetic images predicted to optimally achieve a specific regional response pattern (i.e. maximize the response in a single region)[18]. In our previous paper, synthetic images from NeuroGen with individual-large models were not only having the region-preferred features, but also demonstrated the ability to discover possible individual-level or regional-level preferences which were not easily seen from the noisy fMRI data and the limited images shown to subjects during fMRI. During optimal image generation, we first identified the top optimal image classes by ordering the indices of the 1000 ImageNet classes based on the average predicted activation of class-representative synthetic images generated from 100 random initializations. After that, the class information was one-hot encoded into a class vector and fixed. The noise vector was sampled from a truncated normal distribution, with a truncation parameter of 0.1 to have better image reality. During the optimization, the gradient flows from the region's response back to the synthetic image and then to the noise vector. To see whether NeuroGen combined with the ensemble models can replicate the previous finding of inter-regional and inter-individual preferences of animal vs human faces in various face areas, we generated the top 10 images for three face regions: OFA, FFA1, and FFA2 using NeuroGen with the linear ensemble models as the encoder. We then compared (via Pearson correlation) the ratio of animal vs human faces in those top 10 images to each region's t-statistic of the observed activity in response to animal vs human faces.

**Statistics and reproducibility.** In Fig. 2, Wilcoxon matched-pairs signed rank test was used to compare each model's accuracy with the accuracy of gold standard individual-20K model. In Fig. 3, Friedman's test was used to compare each model's accuracy with every other model's accuracy. FDR method of Benjamini and Hochberg was used to correct for multiple comparisons. In Fig. 4, the one-tailed $p$-values were calculated based on the permutation test and were Bonferroni corrected. The number of pairs was 8 for all NSD related statistical tests and 6 for all NeuroGen related statistical tests. Statistical differences were considered significant if $p < 0.05$. All $p$-values reported in the paper are two-tailed unless otherwise stated.

**Reporting summary.** Further information on research design is available in the Nature Portfolio Reporting Summary linked to this article.

## Data availability

The Natural Scene Dataset is publicly available at http://naturalscenesdataset.org. The NeuroGen Dataset will be made available upon reasonable request. The source data behind Figs. 2–5 are provided in Supplementary Data 1–4.

## Code availability

Code is available at https://github.com/zijin-gu/linear-ensemble[47].

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

## Acknowledgements

This work was funded by the following grants: R01 NS102646 (A.K.), RF1 MH123232 (A.K.), R01 LM012719 (M.S.), R01 AG053949 (M.S.), NSF CAREER 1748377 (M.S.), NSF NeuroNex Grant 1707312 (M.S.), and Cornell/Weill Cornell Intercampus Pilot Grant (A.K. and M.S). The NSD data were collected by Kendrick Kay and Thomas Naselaris under the NSF CRCNS grants IIS-1822683 and IIS-1822929.

## Author contributions

A.K. and M.S. conceived the experiments and interpreted the results, Z.G. conducted the experiments, analysed and interpreted the results. K.J. processed the imaging data and interpreted the results. Z.G. and A.K. wrote the manuscript. All authors reviewed the manuscript.

## Citation gender diversity statement

Recent work in several fields of science has identified a bias in citation practices such that papers from women and other minorities are under-cited relative to the number of such papers in the field[46]. Here we sought to proactively consider choosing references that reflect the diversity of the field in thought, form of contribution, gender, and other factors. We obtained predicted gender of the first and last author of each reference by using databases that store the probability of a name being carried by a woman[46]. By this measure (and excluding self-citations to the first and last authors of our current paper), our references contain 5.12% woman(first)/woman(last), 26.37% man/woman, 15.48% woman/man, and 53.02% man/man. This method is limited in that (a) names, pronouns, and social media profiles used to construct the databases may not, in every case, be indicative of gender identity, and (b) it cannot account for intersex, non-binary, or transgender people. We look forward to future work that could help us to better understand how to support equitable practices in science.

## Competing interests

The authors declare no competing interests.
