## [Peer Review File · Communications Biology]

Reviewers' comments:

Reviewer #1 (Remarks to the Author):

This study proposes a simple method for fitting an individual-subject encoding model given limited data: modeling an individual subject's response vector as a linear combination of the predicted response vectors of other subjects, for whom more data is available. The study evaluates the accuracy of the resulting encoding model, its training-data efficiency, and the extent the resulting model indeed retains subject-specific variability. While the results do not demonstrate an advantage of the proposed approach over simple averaging of the other subjects in terms of accuracy, they indicate that the linear ensemble approach captures significant individual variability in the image-to-response function.

I find the study to be well motivated and the proposed solution elegant. Below, I list a few aspects that, in my opinion, can still be improved.

Major:

1. This work uses non-linear encoding models: both the neural network weights and the linear coefficients that map the activations to the voxels are adjusted during training. Conceptually, non-linear encoding models are very different from linearized encoding models. In linearized encoding models, the neural network weights are frozen and only the linear coefficients are learned. Linearized encoding models are used in the vast majority of current research on artificial neural networks as models of neuronal response since they allow asking whether the training of the neural network has converged to a representation similar to the neuronal representation. Non-linear encoding models (as those used here) allow asking only a much more modest question: does the architecture have sufficient capacity to approximate the biological image-to-response function?

The paper, as it is, glosses over this distinction. It becomes evident that we are dealing with non-linear encoding models only after reading the Methods section. I'm concerned that some readers might misinterpret this paper, especially since most of the cited literature uses linearized encoding models.

Besides making this distinction explicitly early in the paper, the authors might want to consider evaluating their linear ensemble approach in the two regimes: linearized and non-linear encoding models. Such analyses would increase the applicability of the proposed method and the relevance of the paper to the CCN research community.

2. If the authors indeed choose to extend their study to linearized encoding models, then the stage would be set for including a direct comparison of their proposed linear ensemble model with the Bayesian approach suggested by Wen, Shi, Chen & Liu (2018, NeuroImage). In my opinion, such a comparison would considerably improve the impact of the paper.

3. (All analyses) It's difficult to judge the accuracy levels reported without lower bounds on the noise ceiling. Two kinds of estimates can be relevant here: (a) Across subjects - predicting each subject's held-out response vector from the across-subject average of the other subjects' held-out response vectors. (b) Within subjects - use the repeated presentations of the images to form two independent replicates of the data (same set of images, different trials) and calculate the correlation between these two replicates (i.e. "split-half correlation").

These noise ceiling estimates would give the readers a sense of how good (or bad) the encoding accuracy is.

4. Subsection 2.4, Comparison of encoding model accuracies in the out-of-distribution NeuroGen dataset": This section includes only NSD-to-NeuroGen generalization accuracy. It's impossible to judge whether these accuracies indicate excellent or poor generalization without within-distribution accuracy estimated by fitting encoding models only within the NeuroGen dataset. Also, a noise ceiling estimate for the NeuroGen dataset would help us understand to what extent the explainable

variance was indeed explained. At the moment, I find this subsection unconvincing.

5. Subsection 2.4: How were the p-values of the second-level correlation calculated? Since the pairs are not statistically independent (both due to correlation between regions and the participation of each subject in multiple pairs), standard parametric or non-parametric significance tests would be inappropriate (too permissive). One potential solution can be to use a random permutation test. For each simulation, sample one random permutation of the subjects and reuse it across pairs and regions to simulate a null hypothesis of no association that doesn't assume IID observations. To see why this is needed, consider a case where two ROIs have identical observations. This data duplication shouldn't reduce the p-value compared to having just a single region.

6. Data sharing: data sharing over openneuro is better than sharing upon reasonable request.

Minor:

1. Abstract: "face area responses" - which area? It should be the Fusiform Face Area.

2. The abstract should communicate the core idea of the study - modeling each subject's predicted response vector as a linear combination of the other subjects' predicted response vectors. Currently, little information on what is being done is conveyed.

3. Figure 1b: the figure can be made clearer by defining n (the number of subjects).

4. Introduction: "prediction consistency calculated by the Pearson correlation between the inter-subject correlation (ISC) of the predicted activations, or the correlation between the predictions of each pair of subjects, and the ISC of measured activations." - This sentence can be made easier to read by eliminating the conjunction, for example: "prediction consistency calculated by the Pearson correlation between the inter-subject correlation (ISC) of the predicted activations (i.e., the correlation between the predictions of each pair of subjects) and the ISC of measured activations."

5. The meaning of the phrase "deeply-sampled" is unclear.

6. The methods section is not explicit about how the linear ensemble weights were fitted: "The linear ensemble weights were trained using randomly selected samples (number equals to train size) from the original 20K training set." Looking at the GitHub code, I can see that it was ordinary least squares (which is a reasonable choice in this regime). This should be stated explicitly, rather than the ambiguous "trained".

Reviewer #2 (Remarks to the Author):

This manuscript by Gu et al. investigates how much training data is needed to fit an encoding model to predict human fMRI visual cortical responses to natural images. It is currently thought that the responses of 10s of thousands of images (tens of hours of recording time) are needed to properly fit a mapping between deep neural network features and fMRI responses. Instead, the results in this work indicate only ~300 images (~40 min of recording time) is needed. The trick is to leverage encoding models from a pool (or ensemble) of previous subjects. These "ensemble" models may also be used to predict inter-subject variability (i.e., small differences in a brain area's stimulus encoding between subjects). These models have the potential to become personalized encoding models---better capturing the differences in perception and behavior between people of different ages, cultures, learning abilities, and professions.

The paper is easily digestible, and I followed the main results. I liked the idea of looking at prediction accuracy vs training size (Fig 3) *and* prediction consistency (Fig 5)---the latter measure/observation is not really talked about in the literature. I think this paper adds meaningful methodological results to the literature, although I do not think it adds new insights into brain function or computation. I would be happy to review a revised version of the manuscript.

Major comments and concerns:

1) A major point of concern is that there is no investigation into why the average/linear ensembles perform better (Fig. 3, yellow/red dots). Indeed, the average ensemble (no training data) performs at the same level as the individual 20k model (Fig. 3, yellow vs purple dots). This almost surely should not be the case with infinite data (the individual N_k model should outperform) due to individual differences. Thus, I suspect the nice performance of the average/linear ensemble models come from a form of regularization to avoid overfitting. For example, the scratch models need to train $\sim 100,000$ weights (to map the $7 \times 7 \times 2048$ features to fMRI response), whereas the linear ensemble needs to train 8 weights (7 subjects + offset) and the average ensemble, none. Thus, the ensemble models do not have the chance to overfit to the targeted subject's data. Although it appears early stopping is used to fit the linear regression (basically gradient descent and taking the best epoch based on prediction performance of validation data), which is related to an L2 penalty, other regularization techniques (e.g., LASSO) are not performed or mentioned. More insight is needed into why the ensemble models perform better. An interesting analysis would be to have 7 (or more) individual 20k models trained on the same individual but with different random initializations...does taking the ensemble of these do as well as the average ensemble (note this cannot be done with ordinary least squares, which is deterministic and will give the same weights each time)?

2) Give more insight into how many subjects are needed in the ensemble. E.g., a plot of prediction accuracy vs. number of subjects in the ensemble. Does this plateau or keep increasing? This helps give more intuition about the ensemble and predicts how many subjects are needed for a desired performance.

3) That the average ensemble has high prediction with no training samples but the linear ensemble eventually achieves the same performance (but not greater) (Fig 1, > 300 training samples) begs the question if the linear ensemble is adding anything new. What do the Betas look like for the linear ensemble...are the weights $1/N$, where N is the number of subjects. In other words, does the linear ensemble become the average ensemble with enough training samples? If they are not the same (suggested by Fig 5 b/c the linear ensemble has prediction consistency), it is curious that the linear ensemble does not perform better than the average ensemble---two different solutions yield the same prediction accuracy. Explain which setting and how this can be the case to the reader. Of potential interest may be to consider a hybrid approach: for small training sizes, a model is more biased to be the average ensemble; for larger training sizes, it becomes the linear ensemble (i.e., the average ensemble is the prior, and the linear ensemble is more of a posterior with more data). It may be easy enough to achieve this by initializing the linear ensemble's weights to be $1/N$.

4) The manuscript has the claim "The linear ensemble model also has good preservation of inter-individual variability" with $\rho=0.34$ (or an explained variance of $\sim 10\%$). To me, this looks like a poor match to the data compared to the Scratch ($\rho=0.52$) and Individual-20k ($\rho=0.61$) models (this is in Fig 5, top row). More needs to be done to put these values in perspective. For example, what is the ISC between brain areas of the same/different individual(s)? What is the ISC between different models (this may also help telling the difference between linear and average ensemble models)? The text in the Fig 5 caption is confusing: are comparisons made between individuals for the *same* brain area only or also different brain areas? I also am concerned that prediction accuracy plays a role in ISC. For example, two subjects that are more predictable may have a higher ISC. This may change how we interpret the results in Fig 5---prediction consistency may just reflect how predictable subjects are. Plotting average ensemble prediction (averaged between the two subjects) vs. ISC - measurement may help shed light on this.

5) Demonstrating simulated NeuroGen is fine. It would be stronger if there was more investigation in the differences between individuals. Fig 6 mainly piggybacks on a previous result from the NeuroGen paper (animal/dog vs human faces). Perhaps more interesting would be to understand the differences between the average vs linear ensemble models (e.g., finding images that maximize a linear ensemble model while minimizing the average ensemble model). Regardless,

three points here:

- i) I recommend increasing the number of images from 10 to at least 25---giving the ratios more discrete intervals.
- ii) It is unclear to me why I see such similar images in Figure 6b-d. For example, there is an individual with red hair that is almost (but not perfectly) identical in 5 different images. This suggests that BigGAN is highly biased or some initialization was not correct---please find out.
- iii) It is unclear to me, after reading main text + caption + methods, how you can synthesize both human and animal/dog faces, given that you fix a one-hot encoding for the class vector c . What class would have both dog + human faces?! More explanation is needed.

6) Presentation. Overall, the figures were clear. However, in Figure 1, the dimensionality of the mappings should be clear. It looks like the linear mapping (top row, blue box) has a small number of weights, whereas the Betas (bottom row, $B_i, 1 B_i, 2 \dots$) appear to comprise a large number of weights. However, the linear mapping has $7 \times 7 \times 2000$ weights whereas the Betas have 8 weights (including offset). I also thought Figure 2 was more secondary to any claim in the paper (i.e., could be in supplementary), and Figure 4 is mostly redundant with Figure 3 (except for b). In Figure 4, I would remove the violin plots because you already display the data points. Figure 5 could be greatly improved with an illustration (akin to the ones in Figure 1) to make it clear how ISC is computed.

I apologize for my long list of comments, but I like the work, and I believe incorporating these comments will make the manuscript very strong.

Response to Reviewers

Title: Personalized visual encoding model construction with small data

**Manuscript Reference Number:
COMMSBIO-22-1610**

Authors:

Zijin Gu

Keith Jamison

Mert Sabuncu

Amy Kuceyeski

Date: September 23, 2022

Message from the Authors

Dear Reviewers,

We thank the reviewers for their constructive comments, which have allowed us to improve the quality of the manuscript. We have addressed the comments and incorporated these valuable suggestions in the current revision; we believe that the result is a strong manuscript that will be of great interest to the neuroscientific community, vision researchers in particular. The updated contents are colored in blue in the revised manuscript to indicate our changes.

We have made many major changes, including additional analyses and validation and changes to the wording in the text. All page and figure numbers in our response are based on the revised manuscript, unless otherwise stated. The page and reference numbers mentioned in the reviewers' comments are kept intact and are based on the original manuscript. The references contained in this reviewer response document are in author-year format for ease of reading, and are listed at the end of this document. Thank you and we look forward to hearing the journal's decision.

Sincerely,

Zijin Gu, Keith Jamison, Mert Sabuncu, Amy Kuceyeski

Response To Reviewer #1

Overall Comments

This study proposes a simple method for fitting an individual-subject encoding model given limited data: modeling an individual subject's response vector as a linear combination of the predicted response vectors of other subjects, for whom more data is available. The study evaluates the accuracy of the resulting encoding model, its training-data efficiency, and the extent the resulting model indeed retains subject-specific variability. While the results do not demonstrate an advantage of the proposed approach over simple averaging of the other subjects in terms of accuracy, they indicate that the linear ensemble approach captures significant individual variability in the image-to-response function.

I find the study to be well motivated and the proposed solution elegant. Below, I list a few aspects that, in my opinion, can still be improved.

Response

We appreciate your careful review and detailed feedback; we have greatly improved the manuscript based on your comments, to which we provide the point-by-point responses below.

Reviewer Comment

1. This work uses non-linear encoding models: both the neural network weights and the linear coefficients that map the activations to the voxels are adjusted during training. Conceptually, non-linear encoding models are very different from linearized encoding models. In linearized encoding models, the neural network weights are frozen and only the linear coefficients are learned. Linearized encoding models are used in the vast majority of current research on artificial neural networks as models of neuronal response since they allow asking whether the training of the neural network has converged to a representation similar to the neuronal representation. Non-linear encoding models (as those used here) allow asking only a much more modest question: does the architecture have sufficient capacity to approximate the biological image-to-response function?

The paper, as it is, glosses over this distinction. It becomes evident that we are dealing with non-linear encoding models only after reading the Methods section. I'm concerned that some readers might misinterpret this paper, especially since most of the cited literature uses linearized encoding models.

Besides making this distinction explicitly early in the paper, the authors might want to consider evaluating their linear ensemble approach in the two regimes: linearized and non-linear encoding models. Such analyses would increase the applicability of the proposed method and the relevance of the paper to the CCN research community.

Response

Thank you for clarifying the distinction between linear and non-linear models, which was not made evident in the previous version of the paper. We interpret the "neural network" you mentioned in the comment as referring to the feature extractor in the encoding model. While building the encoding models using all available data (individual-20K), we indeed tried two different approaches: 1) freeze the weights of the feature extractor pre-trained from ImageNet (linear); 2) finetune the weights of the feature extractor pre-trained from ImageNet (non-linear). Empirically, we found that finetuning gave better test-time performance than fixing the feature extractor weights. We have added the comparison to the Supplementary Materials and also attach it in the below Figure 1:

Figure S3: Comparison of linear encoding model (ImageNet feature extractor with fixed, pre-trained weights) and non-linear encoding model (ImageNet feature extractor weights were finetuned on the individual's data).

Since the focus of this work is on the creation of an accurate encoding model for a novel individual, and doesn't aim to address the kinds of questions brought up in this comment, i.e. comparison of the artificial and biological neural networks. We have added the following text to the results section regarding this issue:

We note that while most encoding models in the literature use fixed, pre-trained feature extractors, we take an approach of fine-tuning the feature extractor (starting from the fixed, pre-trained weights) in order to achieve a more accurate encoding model (see Supplemental Figure S3). This approach may change the interpretation of the model, but model inference was not the aim of this work - rather it was to provide a way to achieve the most accurate encoding models for a prospective individual using small amounts of data. An exhaustive comparison of the feature extractors of the encoding models used here to neural representations will be the object of future work."

Reviewer Comment

2. If the authors indeed choose to extend their study to linearized encoding models, then the stage would be set for including a direct comparison of their proposed linear ensemble model with the Bayesian approach suggested by Wen, Shi, Chen & Liu (2018, NeuroImage). In my opinion, such a comparison would considerably improve the impact of the paper.

Response

Thank you for this comment and mentioning this important, related previous work. Our finetuned encoding models presented in the manuscript share many similarities with the Bayesian approach used by Wen et al. (2018), with one important difference being that Wen et al. (2018) added an extra constraint in the loss function to penalize the weights of the feature to activation part of the encoding model to be similar to one other subject. Our approach, on the other hand, does not penalize a model that is different from other individuals' models, as we hypothesize that this may reduce the individuality of the encoding models that is currently a strength of our approach. While Wen et al. (2018) didn't explicitly test whether the individual variability was maintained in the encoding models, from the form of the loss function we conjecture that individuality would be minimized. These points are summarized in the introduction of the current version of the paper.

As we illustrated in the above comment's response, finetuning the feature extractor's weights (using non-linear models) leads to better prediction performance. Based on this observation, we decided to use the non-linear approach for the main text. In light of this decision, and other reasons listed below, we believe a fair comparison with the Wen et al. (2018) model is not possible.

1) The experimental setups and data used to train the models are not comparable. It is not straightforward to compare models trained on datasets derived using different experimental setups, fMRI preprocessing steps, visual stimuli (video vs still images), and subjects.

2) The neural activity to be predicted is at a different scale. Wen et al. trained voxel-level encoding models but our focus is on region-level encoding models. A direct comparison of voxel-level results to region-level results would be difficult to achieve as they have very different signal-to-noise properties.

Reviewer Comment

3. (All analyses) It's difficult to judge the accuracy levels reported without lower bounds on the noise ceiling. Two kinds of estimates can be relevant here: (a) Across subjects - predicting each subject's held-out response vector from the across-subject average of the other subjects' held-out response vectors. (b) Within subjects - use the repeated presentations of the images to form two independent replicates of the data (same set of images, different trials) and calculate the correlation between these two replicates (i.e. "split-half correlation").

These noise ceiling estimates would give the readers a sense of how good (or bad) the encoding accuracy is.

Response

We do agree that noise ceiling estimates provide an independent gauge of encoding model performance. Thus, we calculated the two noise ceiling estimates suggested in this comment. We added a subsection in the Methods to explain how we calculated the two noise ceilings:

We calculated two types of noise ceilings (NC) with which to compare our encoding models' prediction accuracies: across-subject NC and within-subject NC. Across-subject NC is computed as the Pearson correlation between subject's measured responses and the average of the other subjects' measured responses across the test data set. Within-subject NC calculations are slightly different across NSD and NeuroGen subjects. For NSD subjects, the within-subject NC is calculated as the Pearson correlation between the average of two measured responses and the third measured response. For NeuroGen subjects, the within-subject NC is calculated as the Pearson correlation between two measured responses in the test set as there are very few images that have three measured responses.

we added these NC values to Figure 4 (see below) along with a description of the metrics in the caption.

Finally, we added some text to the results regarding the noise-ceiling comparisons for the NSD dataset

The across-subject NC, higher than the within-individual NC likely due to a better signal-to-noise ratio, behaved as an upper bound for the prediction accuracy and was always significantly higher than scratch and finetuned models. The individual-20K model only showed significantly lower accuracies than the across-subject NC upper bound for one region (EBA); similarly, the ensemble models also only showed significantly lower accuracies to this upper bound in PPA. The within-subject NC was always significantly lower than the individual-20K and the ensemble model accuracies.

and the NeuroGen dataset.

The ensemble models did not have significantly lower accuracy compared to either NC for any model (and in fact had significantly higher accuracy compared to the within-subject NC for EBA), while scratch and finetuned models had significantly lower accuracies than the across-subject NC in PPA.

Reviewer Comment

4. Subsection 2.4, Comparison of encoding model accuracies in the out-of-distribution NeuroGen dataset": This section includes only NSD-to-NeuroGen generalization accuracy. It's impossible to judge whether these accuracies indicate excellent or poor generalization without within-distribution accuracy estimated by fitting encoding models only within the NeuroGen dataset. Also, a noise ceiling estimate for the NeuroGen dataset would help us understand to what extent the explainable variance was indeed explained. At the moment, I find this subsection unconvincing.

Figure 4: Encoding model prediction accuracies for the 8 NSD individuals **a** FFA1, **b** EBA, **c** PPA, **d** V1v, and 6 NeuroGen subjects **e** FFA1, **f** EBA, **g** PPA, **h** V1v, measured via Pearson’s correlation between predicted and observed regional activity across a set of test images for 4 regions - FFA1, EBA, PPA and V1v. Individual-20K models are illustrated in purple, scratch models in blue, finetuned in green, linear ensemble in orange and average ensemble in yellow. Scratch, finetuned and linear ensemble models were trained with a dataset of 300 image-response pairs from the novel subject in question. The across-subject noise ceiling (NC), representing the correlation between the average measured test-set responses of the other subjects with the measured test-set responses of the subject in question, and within-subject NC, representing the correlation of the individual’s measured image responses, are shown in black and brown, respectively. Black bars at the top of the panels indicate significant differences in accuracy across the indicated pairs of models (Friedman’s test with FDR corrected $p < 0.05$). The correspondence between the number of asterisks and the p -value: * - $p \leq 0.05$, ** - $p \leq 0.01$, *** - $p \leq 0.001$, **** - $p \leq 0.0001$.

Response

We apologize for any confusion; we did include the accuracies from models fitted only using NeuroGen data (the NeuroGen's scratch models). To clarify, the scratch models have the same architecture as the encoding model shown in Figure 1a, are trained only using that subject's data and thus represent within-distribution accuracy. We also have added the noise ceiling estimates for NeuroGen analysis in Figure 4, as described in previous response.

Reviewer Comment

5. Subsection 2.4: How were the p-values of the second-level correlation calculated? Since the pairs are not statistically independent (both due to correlation between regions and the participation of each subject in multiple pairs), standard parametric or non-parametric significance tests would be inappropriate (too permissive). One potential solution can be to use a random permutation test. For each simulation, sample one random permutation of the subjects and reuse it across pairs and regions to simulate a null hypothesis of no association that doesn't assume IID observations. To see why this is needed, consider a case where two ROIs have identical observations. This data duplication shouldn't reduce the p-value compared to having just a single region.

Response

The second-level p-values reported in the manuscript were calculated from Beta distribution, and we do agree that in this context random permutation test is more appropriate. We have thus updated the results in Figure 5 (and associated text) and have added explanation in the Method section about how the p-values are computed. There are no differences in terms of significance levels between the old and new permutation-based p-values.

The updated Figure 5 is attached below.

Reviewer Comment

6. Data sharing: data sharing over openneuro is better than sharing upon reasonable request.

Response

Thank you for this suggestion. We definitely agree that making the dataset/code publicly available would be better for the research community but we are unable to do this under the study's current Institutional Review Board (IRB) for Human Participant Research. We are working to resolve this issue.

Reviewer Minor Comment

Figure 5: Preservation of inter-individual differences within the encoding models. **a** Inter-subject correlations (ISC) are computed as the Pearson correlation between every pair of subjects' predicted (ISC-prediction) or measured (ISC-measurement) brain responses, calculated for each of four brain regions separately. The scatter plots show prediction consistency, which is the relationship between the ISC-measurement and ISC-prediction values across every pair of subjects for the images in the test set; the same region's measured/predicted activity is used for both subjects. The number of points in each scatter plot are therefore the number of brain regions (4) * number of pairs of individuals (for NSD this is $(7 \times 8)/2 = 23$ and for NeuroGen this is $(6 \times 5)/2 = 15$). The scratch, finetuned, and linear ensemble models here were created using a training dataset of 300 image-response pairs for NSD subjects. **b**, **c**, **d** and **e** represent the NSD subjects while **f**, **g** and **h** represent the NeuroGen subjects. Individual-20K models are not available for the NeuroGen individuals. The predicted responses have better SNR than the measured responses (due to the noise in the fMRI responses that is smoothed in the encoding models), thus note the x and y-axis ranges are quite different. All p values (calculated via permutation testing) are one-tailed and Bonferroni corrected for multiple comparisons.

1. Abstract: “face area responses” - which area? It should be the Fusiform Face Area.

Response

We investigated 3 face areas (not all fusiform). We have edited the sentence in the abstract to hopefully increase clarity:

Additionally, we show that the ensemble encoding models are able to discover the inter-individual differences in various face areas’ responses to images of animal vs human faces using a recently developed NeuroGen framework.

Reviewer Minor Comment

2. The abstract should communicate the core idea of the study - modeling each subject’s predicted response vector as a linear combination of the other subjects’ predicted response vectors. Currently, little information on what is being done is conveyed.

Response

Thank you for this comment. We agree that the core idea should be conveyed clearly in the abstract. We have modified the sentence for introducing the linear ensemble model to

Here, we propose an ensemble approach to create encoding models for novel individuals with relatively little data by modeling each subject’s predicted response vector as a linear combination of the other subjects’ predicted response vectors.

Reviewer Minor Comment

3. Figure 1b: the figure can be made clearer by defining n (the number of subjects).

Response

Thank you for this comment. We have added " $n = 7$ or 8 " to Figure 1b.

Reviewer Minor Comment

4. Introduction: “prediction consistency calculated by the Pearson correlation between the inter-subject correlation (ISC) of the predicted activations, or the correlation between the predictions of each pair of subjects, and the ISC of measured activations.” - This sentence can be made easier to read by eliminating the conjunction, for example: “prediction consistency calculated by the Pearson correlation between the inter-subject correlation (ISC) of the predicted activations (i.e., the

correlation between the predictions of each pair of subjects) and the ISC of measured activations.”

Response

Thank you for this suggestion. We have modified the sentence according to your example.

Reviewer Minor Comment

5. The meaning of the phrase “deeply-sampled” is unclear.

Response

Thank you for pointing this out. The phrase "deeply-sampled" as used in our manuscript shares the same meaning as "densely-sampled", which means the dataset contains large scale stimuli-fMRI response pairs. We have made the wording constant in the manuscript by changing "deeply-sampled" to "densely-sampled", and, further, have added a short explanation of the term densely-sampled "i.e. large amounts of data collected from a single individual" to further clarify our meaning.

Reviewer Comment

6. The methods section is not explicit about how the linear ensemble weights were fitted: “The linear ensemble weights were trained using randomly selected samples (number equals to train size) from the original 20K training set.” Looking at the GitHub code, I can see that it was ordinary least squares (which is a reasonable choice in this regime). This should be stated explicitly, rather than the ambiguous "trained".

Response

We agree that it is better to state

The linear ensemble weights were fitted via ordinary least squares using randomly selected samples (number equals to train size) from the original 20K training set.

Response To Reviewer #2

Overall Comments

This manuscript by Gu et al. investigates how much training data is needed to fit an encoding model to predict human fMRI visual cortical responses to natural images. It is currently thought that the responses of 10s of thousands of images (tens of hours of recording time) are needed to properly fit a mapping between deep neural network features and fMRI responses. Instead, the results in this work indicate only 300 images (40 min of recording time) is needed. The trick is to leverage encoding models from a pool (or ensemble) of previous subjects. These "ensemble" models may also be used to predict inter-subject variability (i.e., small differences in a brain area's stimulus encoding between subjects). These models have the potential to become personalized encoding models—better capturing the differences in perception and behavior between people of different ages, cultures, learning abilities, and professions.

The paper is easily digestible, and I followed the main results. I liked the idea of looking at prediction accuracy vs training size (Fig 3) *and* prediction consistency (Fig 5)—the latter measure/observation is not really talked about in the literature. I think this paper adds meaningful methodological results to the literature, although I do not think it adds new insights into brain function or computation. I would be happy to review a revised version of the manuscript.

Response

We would like to thank you for your positive feedback. Your detailed comments have considerably helped to improve the quality of the revised manuscript. We hope you find our below responses satisfactory.

Reviewer Comment

1) A major point of concern is that there is no investigation into why the average/linear ensembles perform better (Fig. 3, yellow/red dots). Indeed, the average ensemble (no training data) performs at the same level as the individual 20k model (Fig. 3, yellow vs purple dots). This almost surely should not be the case with infinite data (the individual N_k model should outperform) due to individual differences. Thus, I suspect the nice performance of the average/linear ensemble models come from a form of regularization to avoid overfitting. For example, the scratch models need to train 100,000 weights (to map the $7 \times 7 \times 2048$ features to fMRI response), whereas the linear ensemble needs to train 8 weights (7 subjects + offset) and the average ensemble, none. Thus, the ensemble models do not have the chance to overfit to the targeted subject's data. Although it appears early stopping is used to fit the linear regression (basically gradient descent and taking the best epoch based on prediction performance of validation data), which is related to an L2 penalty, other

regularization techniques (e.g., LASSO) are not performed or mentioned. More insight is needed into why the ensemble models perform better. An interesting analysis would be to have 7 (or more) individual 20k models trained on the same individual but with different random initializations...does taking the ensemble of these do as well as the average ensemble (note this cannot be done with ordinary least squares, which is deterministic and will give the same weights each time)?

Response

Thank you for this comment. We agree with your thoughts that perhaps the ensemble models perform as well as the individual-20K models is because they don't need to fit such a large number of parameters. However, we believe it is that the individual-20K models are not overfitted; our evidence is the below figure as an example of the learning curve for the individual-20K model. You can see that the training loss and the validation loss initially decrease and then plateau. If the model was overfitting, the training loss would continue to decrease while the validation loss would begin to increase.

An example learning curve for the individual-20K model.

Based on your suggestion, we created 7 individual-20K models trained on the same individual but with different random initializations. We then took the average of the 7 predictions and calculated the prediction accuracy for this "deep-ensemble" approach. The results of this approach and the comparisons with other models are shown in the below figure, with individual-20K ensemble in light blue.

Compared to a single individual-20K model, taking 7 individual-20K models and performing deep ensembling does increase these models' accuracies. However, the boosted accuracy for individual-20K ensemble models has no significant difference comparing with a single individual-20K model, and linear/average ensemble models.

We think the reason why individual-20K models don't perform better than linear/average ensemble is that both types of model have reached the accuracy upper limit, evidenced by the similar accuracy when comparing with the across-subject noise ceiling (NC), which is calculated as the Pearson correlation between the fMRI response for the query subject and the averaged fMRI response across all other subjects.

Figure: Model accuracy on FFA1 comparison with an additional deep-ensembled individual-20K model (average of 7 individual-20K models trained with varied initializations) for all 8 NSD subjects.

Reviewer Comment

2) Give more insight into how many subjects are needed in the ensemble. E.g., a plot of prediction accuracy vs. number of subjects in the ensemble. Does this plateau or keep increasing? This helps give more intuition about the ensemble and predicts how many subjects are needed for a desired performance.

Response

This is certainly an important piece of information. We have now plotted the relationship between prediction accuracy and the number of pretrained encoding models used in linear ensemble in the below Figure S4 a-d for FFA1, EBA, PPA and V1v in NSD subjects and f-i in NeuroGen subjects, respectively. As shown in the figure, besides an obvious boost of accuracy from 1 to 2 pretrained models in the linear ensemble for NSD subjects, the prediction accuracies are very similar when using a different number of subjects' pretrained encoding models in the linear ensemble model. However, we observed a positive relationship between the number of subjects' pretrained encoding models used and the prediction consistency score (Spearman's $r = 0.96$, $p = 4.54e - 4$ for NSD and Spearman's $r = 0.93$, $p = 8.63e - 4$ for NeuroGen). This indicates that the individual variability is not preserved when using a small number of training subjects, e.g. one to three, as the predictions would be extremely biased towards the training subjects which may not be very representative of the novel individual. Further, the prediction consistency score is similar for 6-8 individuals.

We have added the following text to the results section:

Finally, we investigated how varying the number of individuals used in the linear ensemble model affected prediction accuracy and consistency, see Figure S4. We see that there is an increase in accuracy from using one to two individuals' models in the linear ensemble for the NSD subjects, but additional individuals beyond two do not further increase accuracy. For NeuroGen, linear ensemble model accuracy does not seem to

Figure S4: Relationship between number of pretrained encoding models included in the linear ensemble approach and **a,b,c,d** prediction accuracy and **e** prediction consistency in NSD subjects; **f,g,h,i** prediction accuracy and **j** prediction consistency in NeuroGen subjects.

increase with increasing number of individuals' models. However, for both datasets, the prediction consistency, or the preservation of inter-individual differences in the predictions, increases with the number of individuals' models used in the linear ensemble. This indicates the importance of using at least several individuals in a linear ensemble in order to maintain inter-individual differences in a linear ensemble approach.

and the following text to the discussion:

Furthermore, we provide quantitatively derived guidelines for how many images are needed to achieve accuracy similar to encoding models trained on very large-scale data in Figure 3, and the effects of varying the number of individuals' pretrained models used in the linear ensemble model in Figure S4. For the latter analysis, we conjecture that, since we are using the higher signal-to-noise ratio (SNR) individual-20K encoding model predictions in the linear ensemble (and not the lower SNR measured fMRI responses), that even using one individual's encoding model gives us the highest prediction accuracy we can get (without the prediction consistency). Thus, adding more individuals' encoding models to the linear ensemble does not increase model accuracy but does increase prediction consistency, i.e. better preserves the inter-individual relationship of responses across individuals.

Reviewer Comment

3) That the average ensemble has high prediction with no training samples but the linear ensemble eventually achieves the same performance (but not greater) (Fig 1, > 300 training samples) begs the question if the linear ensemble is adding anything new. What do the Betas look like for the linear ensemble...are the weights $1/N$, where N is the number of subjects. In other words, does the linear ensemble become the average ensemble with enough training samples? If they are not the same (suggested by Fig 5 b/c the linear ensemble has prediction consistency), it is curious that the linear ensemble does not perform better than the average ensemble—two different solutions yield the same prediction accuracy. Explain which setting and how this can be the case to the reader. Of potential interest may be to consider a hybrid approach: for small training sizes, a model is more biased to be the average ensemble; for larger training sizes, it becomes the linear ensemble (i.e., the average ensemble is the prior, and the linear ensemble is more of a posterior with more data). It may be easy enough to achieve this by initializing the linear ensemble's weights to be $1/N$.

Response

Thank you for this comment. In fact, we see that the weights for linear ensemble models are not close to a the average model's $1/N$ weights (new supplementary Figure S5):

we have added the following text in reference to this figure in the results:

Figure S5 further visualizes the linear ensemble model coefficients across all subjects and brain regions; we can see that there is wide variability in the weights for the linear ensemble models over the individual and brain region in question.

a NSD subjects (in-distribution)

b NeuroGen subjects (out-of-distribution)

Figure S5: Weights for the linear ensemble models for **a** all NSD subjects and **b** all NeuroGen subjects for all brain regions.

We have now added some text to the discussion as to why we think that the linear ensemble does not perform better than the average ensemble:

Prediction accuracy is the most common metric when evaluating neural encoding models. However, few works have investigated how the models preserve inter-individual variability. For example, a model only predicting the population average (or just using responses from one other individual) may have high prediction accuracy, however these models do not preserve individuality of a subject's brain responses. This work focused on evaluating both accuracy and individuality (here called consistency) with the aim to create models that are both accurate and personalized. We generally found a trade-off between prediction accuracy and consistency, which may explain why the linear ensemble models (which maintain consistency) did not outperform the average ensemble models (which do not maintain consistency).

Reviewer Comment

4) The manuscript has the claim "The linear ensemble model also has good preservation of inter-individual variability" with $\rho=0.34$ (or an explained variance of 10%). To me, this looks like a poor match to the data compared to the Scratch ($\rho=0.52$) and Individual-20k ($\rho=0.61$) models (this is in Fig 5, top row). More

needs to be done to put these values in perspective. For example, what is the ISC between brain areas of the same/different individual(s)? What is the ISC between different models (this may also help telling the difference between linear and average ensemble models)? The text in the Fig 5 caption is confusing: are comparisons made between individuals for the *same* brain area only or also different brain areas? I also am concerned that prediction accuracy plays a role in ISC. For example, two subjects that are more predictable may have a higher ISC. This may change how we interpret the results in Fig 5—prediction consistency may just reflect how predictable subjects are. Plotting average ensemble prediction (averaged between the two subjects) vs. ISC - measurement may help shed light on this.

Response

Thank you for this opportunity to increase clarity and significance of the results. Because the scratch and individual-20K models are directly and solely trained on the individual’s data, they are well poised to capture the subject’s individuality. However, due to the limited number of training samples for the scratch model, it cannot achieve good prediction accuracy (as shown in Figure 4). Since individual-20K models have access to large scale individual data, they perform well on both prediction accuracy and consistency. We think it is reasonable that there is a drop of prediction consistency for the linear ensemble models compared with the above two types of models. This work’s emphasis is on the linear ensemble’s balance of higher accuracy (compared to scratch) and increased consistency (compared to the average ensemble).

We have added the following description of a lower bound for the consistency metric to the results section:

The average ensemble model does not preserve inter-individual variability in predictions and thus can serve as a lower-bound for comparing other models’ consistency values. Since the ISCs of the average ensemble model’s predicted activities will all be 1 or near 1 (with leave-one-out training), the prediction consistency of the average ensemble models is ~ 0 or undefined.

All the scatter plots in Figure 5 include all four regions (FFA1, EBA, PPA and V1v) that we modelled. The number of points is thus the number of brain regions (4) * the number of unique pairs of subjects (for NSD this is $(7 \times 8)/2 = 23$ and for NeuroGen this is $(6 \times 5)/2 = 15$); the prediction consistency was calculated based on all of these values of ISC. We have thus edited the caption of Figure 5 to contain this information clarity of interpreting the figure:

Preservation of inter-individual variability within the encoding models. The scatter plots show prediction consistency, which is the relationship between the inter-subject correlations (ISC) of measured activity and inter-subject correlations (ISC) of predicted activity. The inter-subject correlations are calculated across every pair of subjects for the images in the test set; the same region’s measured/predicted activity is used for both subjects. The number of points in each scatter plot are therefore the number of brain regions (4) * number of pairs of individuals (for NSD this is $(7 \times 8)/2 = 23$ and for NeuroGen this is $(6 \times 5)/2 = 15$). The scratch, finetuned, and linear ensemble models here were created using a training dataset of 300 image-response pairs for NSD subjects. **a**, **b**, **c** and **d** represent the NSD subjects while **e**, **f** and **g** represent the NeuroGen subjects. Individual-20K models are not available for the NeuroGen individuals. All p

values (calculated via permutation testing) are one-tailed and Bonferroni corrected for multiple comparisons.

We did observe a positive relationship between linear ensemble prediction accuracy and ISC-measurement, as shown in the below figure. However, this result is not surprising - if there is a

Figure: Relationship between average linear ensemble model accuracy across pairs of the subjects and the ISC of those subjects' measurements.

subject that is very similar to another (ISC-measurement is high) then an ensemble model for one of these individuals that includes the other (similar) individual will also be accurate (and vice versa). If a subject has low ISC with the other individuals, for example, then an ensemble model based on their predicted values will not be accurate. Thus, the prediction accuracy and consistency are not totally independent of each other (but also not representing exactly the same value).

Reviewer Comment

5) Demonstrating simulated NeuroGen is fine. It would be stronger if there was more investigation in the differences between individuals. Fig 6 mainly piggybacks on a previous result from the NeuroGen paper (animal/dog vs human faces). Perhaps more interesting would be to understand the differences between the average vs linear ensemble models (e.g., finding images that maximize a linear ensemble model while minimizing the average ensemble model). Regardless, three points here: i) I recommend increasing the number of images from 10 to at least 25—giving the ratios more discrete intervals. ii) It is unclear to me why I see such similar images in Figure 6b-d. For example, there is an individual with red hair that is almost (but not perfectly) identical in 5 different images. This suggests that BigGAN is highly biased or some initialization was not correct—please find out. iii) It is unclear to me, after reading main text + caption + methods, how you can synthesize both human and animal/dog faces, given that you fix a one-hot encoding for the class vector c . What class would have both dog + human faces?! More explanation is needed.

Response

We agree that finding images that maximize a linear ensemble model while minimizing the average ensemble model would be an interesting direction, but it is not the main focus of the current paper. For this work, we focus on demonstrating that NeuroGen maintains the ability to discover and

amplify individual response differences when using our small-data trained linear ensemble encoding model for a novel individual. This is an important step for prospective experiments that plan to utilize the NeuroGen framework, which we discuss in the paper.

With regards to your itemized comments: i) We show only the top 10 images as we think it is a stronger argument that the individual differences can be reflected and amplified using only 10 images from NeuroGen. We did extend the number of images to 25, and the results are nearly identical, with the correlation between the fMRI t-statistic and the animal vs. human face ratio in top 25 images being Pearson's $r = 0.5294$ and $p = 0.024$. We do include the top 25 images results in the Supplementary Material (see Figure S6).

ii/iii) In Figure 6b-d, we are displaying the top 10 images for OFA, FFA1 and FFA2, all of which are face perception areas. To clarify, a different one-hot encoded class vector for each of the top 10 images was used, but across subjects or regions there may be the same class vector resulting in similar looking images across subjects or regions. Thus, it is not surprising that there are similar images (of faces) appearing in the three figures. The images of an individual with red hair are generated using the same class vector (across different regions/subjects) but have different noise vectors, therefore they look similar but are not identical.

We have added the following text to the results section to clarify this:

Note: each of the top 10 images uses a different one-hot encoded class vector but some subjects/regions top 10 image sets may have the same class vector leading to similar-looking images across subjects/regions top 10 image sets (for example, the person wearing a red wig).

Reviewer Comment

6) Presentation. Overall, the figures were clear. However, in Figure 1, the dimensionality of the mappings should be clear. It looks like the linear mapping (top row, blue box) has a small number of weights, whereas the Betas (bottom row, $B_{i,1}$ $B_{i,2}$...) appear to comprise a large number of weights. However, the linear mapping has $7 \times 7 \times 2000$ weights whereas the Betas have 8 weights (including offset). I also thought Figure 2 was more secondary to any claim in the paper (i.e., could be in supplementary), and Figure 4 is mostly redundant with Figure 3 (except for b). In Figure 4, I would remove the violin plots because you already display the data points. Figure 5 could be greatly improved with an illustration (akin to the ones in Figure 1) to make it clear how ISC is computed.

Response

1) The squares in Figure 1 are just illustrations of the weights and the length actually doesn't represent any information about the dimensionality. For clarification, we added the exact number of encoding models used (the value of n) to the figure.

2) Figure 2 is not the primary result of the paper but we believe it is important to show the shape of the curve as the number of image-response data pairs increases as this may be used to inform prospective studies' experimental designs (i.e. how many image-measured responses are needed to

achieve good accuracy).

3) We decided to keep the violin plots (with median and quartiles) and the data points together but removing the connected lines between points to simplify the figure. The violin plot is a summary of the distribution but we can also see the individual points. We hope this will make the figure more clear.

4) We have added an illustration of the ISC calculation to Figure 5, shown here.

Figure: **a** Inter-subject correlations (ISC) are computed as the Pearson correlation between every pair of subjects' predicted or measured brain responses, calculated for each of four brain regions separately.

Response To Reviewer #3

Overall Comments

In this paper Gu and colleagues build encoding models for the recently published Natural Scenes Dataset and a new dataset the NeuroGen dataset. The goal of this paper is to build ‘personalized encoding models’ – encoding models that predict data for new participant’s personalized data using as little subject-specific data as possible. The goal of personalized encoding model is really is to preserve as much as of the subject-specific response to stimuli as possible. The central claim of the paper is that ensemble models trained on data from other individuals can predict personalized data from the new individual. If true, this claim could be helpful in real-world application of this computational framework. However, the current data are insufficient to back up this central inference. Below I review the evidence in support of the central claim first and before discussing other potential methodological concerns.

Response

Reviewer Comment

Central claim: The authors should at the outset define what they mean exactly by ‘personalized’ given that this is the central claim of this study. Per the definition of personalized, the authors’ claim would imply the ability to predict the variance unique to a specific individual. To then demonstrate this empirically, it is imperative to first demonstrate convincingly that there is indeed some replicable personalized (unique) subject-specific variance in the data to be predicted in the first place, and then show success at predicting that variance. I believe that the most pertinent section and figure in support of the central claim of the paper are Section 2.4 and Figure 5 respectively.

Response

Our central claim in this paper is that linear ensemble models can have a good balance between prediction accuracy (demonstrated in Figure 4) and the ability to preserve the inter-individual similarity/differences of measured responses (demonstrated in Figure 5). We agree that the word "personalized" was ambiguous; we used this term to indicate the model’s ability to preserve the inter-individual similarity/differences, not the ability to predict the variance unique to a specific individual as you mentioned. We added some text in the introduction to clarify our meaning of the term "personalized":

Besides good accuracy, we also aim to make the model personalized, i.e. demonstrate the model’s ability to preserve the inter-individual similarity/differences of measured responses.

Figure S1: Top 10 NSD images that have the highest fMRI measured brain responses in each of the 4 brain regions (FFA1, EBA, PPA, V1v) for all 8 subjects.

We also changed every instance of "inter-individual variability" to "inter-individual differences" to emphasize we are not discussing on individual's variance from the population mean.

Our results do demonstrate a robust and replicable presence of inter-individual differences brain responses. For example, the top 10 NSD images that have the highest fMRI *measured* activations in each of the 4 brain regions for all subjects are shown in Figure S1 and also attached below. Although there are some common top images across subjects, there is also quite a bit of variability. Of course, the biggest evidence for inter-individual differences is the ISC-measurement: if all subjects have the same responses (plus or minus some error term), then the ISC-measurement will be closer to 1 for pairs of subjects. However, the ISC-measurement has a large range (0.2 – 0.8 for NSD and –0.2 – 0.6 for NeuroGen), indicating that there are inter-subject differences. Finally, in response to this and other comments, we tested the reliability of the ISC-measurement and found it was relatively robust, see our response to your third comment below for more details.

We added the following text to the Result section:

We first demonstrated that there are reliable inter-individual differences in brain responses, evidenced by the low ISC-measurement values in Figure 5 and the variability of the top images for NSD subjects shown in Supplementary Figure S1.

Reviewer Comment

1. Here the authors show that the linear ensemble model, the model that has access to some part of the held-out individual's data, is better than the scratch model. Conceptually this is not surprising, but I had a hard time following Figure 5 which must be re-done. For instance, what does each dot indicate? Shouldn't these be done per-region? The authors should also show plot these scatterplots after equating the x and y-axes and clearly indicating the $x=y$ line on these plots. These changes will aid readers' ability to clearly see the differences, if any, between predicted and measured ISC measure.

Response

The scatter plots in Figure 5 show prediction consistency, which is the relationship between the inter-subject correlations (ISC) of measured activity and inter-subject correlations (ISC) of predicted activity. We have now added a panel to Figure 5 to illustrate how ISC is computed (per region) for clarity. We agree that the content of the scatter plots in Figure 5 was not made clear originally - we have updated the text and captions to better describe its content. Specifically, all regions and all pairs of individuals are plotted here - thus the number of points in each scatter plot are thus the number of brain regions (4) * number of pairs of individuals (for NSD this is $(7 \times 8)/2 = 23$ and for NeuroGen this is $(6 \times 5)/2 = 15$).

Given the noisy nature of the fMRI responses and the encoding model's smoothing nature, it is unsurprising that the ISC-prediction is generally higher than the ISC-measurement. Adding the identity line would hinder the clarity of the figure; we have added some text to the caption to highlight this issue.

The updated Figure 5 is attached below.

Reviewer Comment

2. The measured ISC represents how much variance is shared between every pair of individuals. If a model is indeed good, the model should also capture the shared variance between individuals. But the linear ensemble model here has the lowest correlation between the predicted and measured ISC, indicating that it does not capture this shared variance. For instance, I see measured ISC values as low as 0.2-0.3 and the predicted ISC values as high as 0.9 (Figure 5d). But the authors conclude that the "linear ensemble model has the best balance of accuracy and preservation of inter-individual differences". It is not clear to me what data supports the inference.

Figure 5: Preservation of inter-individual differences within the encoding models. **a** Inter-subject correlations (ISC) are computed as the Pearson correlation between every pair of subjects' predicted (ISC-prediction) or measured (ISC-measurement) brain responses, calculated for each of four brain regions separately. The scatter plots show prediction consistency, which is the relationship between the ISC-measurement and ISC-prediction values across every pair of subjects for the images in the test set; the same region's measured/predicted activity is used for both subjects. The number of points in each scatter plot are therefore the number of brain regions (4) * number of pairs of individuals (for NSD this is $(7 \times 8)/2 = 23$ and for NeuroGen this is $(6 \times 5)/2 = 15$). The scratch, finetuned, and linear ensemble models here were created using a training dataset of 300 image-response pairs for NSD subjects. **b**, **c**, **d** and **e** represent the NSD subjects while **f**, **g** and **h** represent the NeuroGen subjects. Individual-20K models are not available for the NeuroGen individuals. The predicted responses have better SNR than the measured responses (due to the noise in the fMRI responses that is smoothed in the encoding models), thus note the x and y-axis ranges are quite different. All p values (calculated via permutation testing) are one-tailed and Bonferroni corrected for multiple comparisons.

Response

As stated in the response to the previous comment, the fMRI measurement noise leads to relatively low ISC-measurement values (range 0.2-0.8 for NSD data) compared to the ISC-predicted which is based on the smoothed predicted responses from the encoding models (range 0.7-1.0 for NSD linear ensemble model). Thus, we do not expect the ISC-measurement and ISC-predictions to be on the $x = y$ line.

The reason we state that the "linear ensemble model has the best balance of accuracy and preservation of inter-individual differences" is because: 1) it has prediction accuracy that is not significantly different from the individual-20K model's accuracy and the across-subject noise ceiling (new result in response to another comment) and 2) it has significantly positive prediction consistency. While the average ensemble model also gives good prediction accuracy, it does not preserve inter-individual variability in predictions since the ISCs of the average ensemble model's predicted activities will all be 1 or near 1 (with leave-one-out training). Further, while the scratch models preserve well the inter-subject differences, the accuracy is lower than the individual-20k models. Therefore, among all the small-data derived models we evaluated, the linear ensemble has the best balance of accuracy and preservation of inter-individual differences.

We include additional text to describe this reasoning in the results section:

These results demonstrate that, out of the four models using small data, the linear ensemble model has the best balance of accuracy and preservation of inter-individual differences. Importantly, it also achieves accuracy similar to the individual-20K model using a training dataset that is only 1.5% the size of the larger model's training data.

Reviewer Comment

3. To get back to the central claim of the paper which is that encoding models capture the personalized response variance. While this analysis in this section attempts to measure the shared variance between subjects I did not find any quantification of replicable subject-specific personalized variance and the ability of the model to predict this unique and replicable subject-specific variance. The other figures and sections in the paper also do not tackle this specific question head-on. There are no estimates of the replicable unique subject-specific personalized variance observed in the data. There is also no quantification of how of this personalized variance is explained by any of the models. Together, I find that current evidence do not support the central claim of the paper.

Response

As we responded to your first comment, our central claim in this paper is that linear ensemble models can have both good prediction accuracy and the ability to preserve the inter-individual differences. The term "inter-individual differences" (as opposed to our previous term "inter-individual variability" which was misleading) has now more clearly defined in the paper to emphasize we are not looking at some statistical measure of an individual's personalized variance but rather preservation of inter-individual differences as measured with our prediction consistency metric.

We have further analyzed the reliability of the ISC-measurement values by recalculating its value using 1000 random subsamples of the data. Below we show 1) the relative error in the ISC-

Figure: **a** The relative error in the ISC-measurements (original value - reshuffled value)/original value, across all pairs of subjects and all regions. **b** The distribution of the individual-20K model's prediction consistency calculated using the 1000 subsamples (original value using all the data is in red).

measurements (original value - reshuffled value)/original value, across all pairs of subjects and all regions and 2) the distribution of the individual-20K model's prediction consistency calculated using the 1000 subsamples (original value using all the data is in red). The ISC-measurement value, which captures the inter-subject differences in measured responses, does seem robust and replicable across different divisions of the data, suggesting that this measure is indeed capturing some true underlying inter-individual differences and not just noise.

We have added this figure to the Supplemental information and have added associated text to the results section of the paper:

We assessed the reliability of the prediction consistency and ISC-measurement metrics by randomly subsampling the data 1000 times and recalculating the ISC-measurement and the individual-20K model's prediction consistency. We indeed see robustness of the ISC-measurement, with the relative errors being mostly within 3% of their original values, as well as the prediction consistency which varied within only 10% of the original value (see Supplementary Figure S8).

Reviewer Comment

4. Given its central importance, it is very difficult to follow all the jargon in this section that switches between ISC consistency, ISC measure, ISCs and inter-individual variability. These are all used, in some cases interchangeably, in this paragraph. I urge the authors to also reconsider all the jargon and make it easier for readers to understand.

Response

We defined in our paper that the prediction consistency, which is the relationship between the inter-subject correlations (ISC) of measured activity (ISC-measurement) and ISC of predicted activity

Figure: Prediction accuracy when using the first ResNet block and the final ResNet block as feature extractor in the encoding model for subject 1.

(ISC-prediction), can be viewed as a measurement of the amount of inter-individual differences that the encoding model preserves. We have modified the related text (particularly the caption of Figure 5) to improve clarity.

Reviewer Other Comment

1. It is unclear why the authors only chose the final layer (presumably the maxpool layer based on Figure 1?) to build encoding models. This is potentially problematic for several reasons. First, The dependence between region depth and model depth has been shown several times and this choice is not optimal especially for V1. Second (more pressing issue) is that this choice could disadvantage the scratch, fine-tuned models more than the ensemble model because this layer may not provide an adequately expressive representational bases to predict the responses. Finally, this choice could also lower the performance of the individual 20K model (which the authors seem to be using as a ceiling, see related next comment).

Response

In the Method section of the paper, we state: "The image feature maps (shape $7 \times 7 \times 2048$) are derived from the final layer of the ResNet50 backbone, as empirically we found that the feature maps derived from later layers are able to predict with good accuracy for both early visual areas and higher order regions."

To give more detail, before deciding the model architecture we conducted an experiment by varying layer depth for image feature extraction. We observed that, for all regions (including late and early visual), models using the deeper layers outperformed models using the shallower layers. We attach below the comparison of the model accuracies by layer depth to support our statement.

We have added some discussion to the methods section to expand our design choice.

Since the weights of the feature extractor derived from pretraining on ImageNet were not fixed but instead finetuned using the neural data, empirically we found that encoding models based on feature maps derived from later layers had better accuracy (compared to earlier layers) for both early visual areas and higher order regions.

Reviewer Other Comment

2. “We consider the individual-20K model as the gold-standard”. The ceiling for a given model should be the replicable variance of the data for a given region, and not this performance of the individual 20K model which is highly dependent on the specific encoding model choice. The authors should quantify and use this as the ceiling for the models and not the performance of their 20K model.

Response

In response to this and other reviewer comments, we have added a within-subject noise ceiling (NC) and between-subject NC with which to compare the encoding models’ accuracies. We added text on how the NCs are calculated.

We calculated two types of noise ceilings (NC) with which to compare our encoding models’ prediction accuracies: across-subject NC and within-subject NC. Across-subject NC is computed as the Pearson correlation between subject’s measured responses and the average of the other subjects’ measured responses across the test data set. Within-subject NC calculations are slightly different across NSD and NeuroGen subjects. For NSD subjects, the within-subject NC is calculated as the Pearson correlation between the average of two measured responses and the third measured response. For NeuroGen subjects, the within-subject NC is calculated as the Pearson correlation between two measured responses in the test set as there are very few images that have three measured responses.

and added them to Figure 4, the updated version of which is provided below.

Reviewer Other Comment

3. Figure 6 would benefit from control measures showing that the animal-person ratio is not trivial. How reliable in the animal-person ratio in the actual data over say splits of the data? How correlated is the animal-person ratio to the data reliability of the fMRI data?

Response

We performed additional analyses quantifying the robustness of the animal-person ratio, and thus added a description of the results to the manuscript:

We tested the reliability of this result by randomly sub-sampling 1000 splits of the NeuroGen data and re-calculating the animal-person t-statistic and its correlation with

Figure 4: Encoding model prediction accuracies for the 8 NSD individuals **a** FFA1, **b** EBA, **c** PPA, **d** V1v, and 6 NeuroGen subjects **e** FFA1, **f** EBA, **g** PPA, **h** V1v, measured via Pearson’s correlation between predicted and observed regional activity across a set of test images for 4 regions - FFA1, EBA, PPA and V1v. Individual-20K models are illustrated in purple, scratch models in blue, finetuned in green, linear ensemble in orange and average ensemble in yellow. Scratch, finetuned and linear ensemble models were trained with a dataset of 300 image-response pairs from the novel subject in question. The across-subject noise ceiling (NC), representing the correlation between the average measured test-set responses of the other subjects with the measured test-set responses of the subject in question, and within-subject NC, representing the correlation of the individual’s measured image responses, are shown in black and brown, respectively. Black bars at the top of the panels indicate significant differences in accuracy across the indicated pairs of models (Friedman’s test with FDR corrected $p < 0.05$). The correspondence between the number of asterisks and the p -value: * - $p \leq 0.05$, ** - $p \leq 0.01$, *** - $p \leq 0.001$, **** - $p \leq 0.0001$.

Figure: For each individual and each face area, we plotted of t-statistic of that individual/region’s measured fMRI responses to dog vs. person faces and that individual/region’s fMRI data reliability. No relationship was identified.

the top 10 synthetic images. We found it to be relatively robust with a coefficient of variation of $\pm 12.3\%$.

We then plotted the relationship between the animal vs. person t-statistic for each person and each face region and an estimate of their data reliability, calculated as the correlation of two repeated response measurements to the same images (also known as within-subject noise ceiling). We found that there is no association between these two variables (Pearson’s $r = -0.04$, $p = 0.88$). The figure is attached below.

Reviewer Other Comment

4. This work builds heavily on the NSD fMRI data which the authors do an exemplary job of making publicly accessible. Given this, the choice of making the new NeuroGen dataset available on request in highly sub-optimal.

Response

We definitely agree that making the NeuroGen dataset publicly available would be better for the research community but we are unable to do this under the study’s current Institutional Review Board (IRB) for Human Participant Research. We are working to resolve this issue.

Reviewer Other Comment

Clarification: According to the authors the validation set was determined as the 766 images from the shared 100 set that had at least 2 presentations across all subjects. Didn't a few of the subjects from NSD have 500 shared images?

Response

The NSD paper Allen et al. (2021) states:

We designated a special set of 1,000 images (chosen randomly from the full set of prepared images) as shared images that would be seen by all participants (referred to as the 'shared1000');

And we stated the reason in the manuscript why we selected 766 images from the shared 1000 as below:

For each NSD subject, we selected the 766 images from the shared 1000 set of images that had at least 2 presentations per subject, over all subjects.

Some images were not seen at least twice by all subjects and were thus excluded. _____

References

- Allen, E. J., St-Yves, G., Wu, Y., Breedlove, J. L., Prince, J. S., Dowdle, L. T., Nau, M., Caron, B., Pestilli, F., Charest, I., et al. (2021). A massive 7t fmri dataset to bridge cognitive neuroscience and artificial intelligence. *Nature neuroscience*, pages 1–11.
- Wen, H., Shi, J., Chen, W., and Liu, Z. (2018). Transferring and generalizing deep-learning-based neural encoding models across subjects. *NeuroImage*, 176:152–163.

Reviewers' comments:

Reviewer #1 (Remarks to the Author):

I thank the authors for substantively addressing my comments about introducing noise ceiling estimates. I believe that the results are more interpretable now.

I still believe that the focus on non-linear encoding models is not well motivated scientifically, but this is the authors' paper.

Regarding the response: "we did include the accuracies from models fitted only using NeuroGen data (the NeuroGen's scratch models)." This is still not evident from the main text and figure and should be explicitly stated. At the moment, the NeuroGen scratch model performance is plotted under the title "NeuroGen Subjects (out-of-distribution)", but if I understand the authors' response correctly, this blue violin plot describes within-distribution performance.

Other than that, I have no further comments.

Reviewer #2 (Remarks to the Author):

I have re-read my comments and read the authors' rebuttal. I have one major concern that needs to be addressed by the authors in rebuttal. I also have minor concerns that should be addressed by the authors, but I do not need a rebuttal/see a revised manuscript before acceptance.

Major concern:

"We did observe a positive relationship between linear ensemble prediction accuracy and ISC measurement, as shown in the below figure. However, this result is not surprising - if there is a subject that is very similar to another (ISC-measurement is high) then an ensemble model for one of these individuals that includes the other (similar) individual will also be accurate (and vice versa)."

I have a major concern with the metric of consistency, as defined by this manuscript. For real data, it is fine to define ISC consistency as the correlation between responses of two subjects across the same images. This metric can accurately identify the presence of inter-individual differences (but not what those differences are).

However, as displayed by the reviewer figures, using this definition of consistency for the models is redundant with prediction power and leads to ambiguity. I'll give a few examples to make this clear.

First, consider the case where the models (e.g., one linear ensemble model per subject) perfectly predict the subjects' responses. It follows that these models would perfectly match in consistency with the real data (i.e., a large rho for ISC-measurement vs ISC-prediction) because the models perfectly predict all responses. Thus, the models are consistent simply because they have perfect prediction. I suppose this makes sense—perfect prediction should mean that the models perfectly identify inter-individual differences.

Second, consider the case where the responses of the first N-1 subjects are identical but the responses of the Nth subject are uncorrelated (i.e., an ISC=0 between the Nth subject and any other subject). Then consider a model that only outputs responses identical to the responses of the first N-1 subjects. This model would also have perfect consistency, because it too would have an ISC=0 when comparing the model for the Nth subject vs any other subject's model. However, it is unable to predict any response for the Nth subject, so it fails to capture any real inter-individual difference. Its high consistency stems solely from the fact that it has high prediction.

Third, we now consider the fMRI data in this study. There is a strong correlation between prediction power and consistency (via the reviewer figures). Is it the case that the encoding models are pulling out true inter-individual differences? Or rather this match in consistency (Figure 5 scatter plots) simply arises because the models are better able to predict some subjects more than others? Arguing that prediction is high enough for all subjects (Figure 4) is difficult because the model could be predicting the portions of responses shared across all subjects. A potential confound is that the models do not pull out true inter-individual differences; instead, the match in consistency is because some models fail to predict the “outlier” subjects.

In the end, I think using the Pearson correlation as a consistency metric is too coarse. *The manuscript needs to convince the reader that the models truly pull out inter-individual differences.* To that end, one idea is to analyze the residuals between two true subjects versus the residuals between two models. Wouldn't taking the correlation between these two residual vectors be a better measure of consistency? This metric would still rely on prediction power in the sense that models with perfect prediction would have $\rho=1$. However, it better eliminates the possibility that a correlation in consistency might be caused because a model cannot predict an outlying subject's responses. A high consistency here indicates that the response differences between subjects are correlated with the response differences between models.

Minor revisions:

“We agree with your thoughts that perhaps the ensemble models perform as well as the individual-20K models is because they don't need to fit such a large number of parameters. However, we believe it is that the individual-20K models are not overfitted”

I apologize, I was using the broad definition of the term “overfitting”. I did not mean that the particular individual-20K models were overfitting due to the optimization procedure or regularization (which you make clear in the training curve figure). Instead, another form of “regularization” is the number of parameters that need to be fit. I'm thinking about the case of fitting a linear line vs a polynomial fit with a large degree—the latter will “overfit” if it does not have enough data. This is one potential cause why the individual-20K model does not achieve better performance.

Regardless, mention to the reader why performance is about the same for the individual-20K vs the average ensemble model (Fig. 3)---your reasoning about hitting the upper bound of prediction makes sense, as well as concerns for overfitting. It is an interesting point to bring up, because one could argue the ensemble itself is prone to overfitting (as it relies on 7 large models). This might be confusing to readers of why this works—it would be helpful to clarify.

“The images of an individual with red hair are generated using the same class vector (across different regions/subjects) but have different noise vectors, therefore they look similar but are not identical.”

The part that tripped me in Figure 6 caption is “The image generator takes in a class vector which is fixed during optimization...”. Change this language to indicate that different images may have different class vectors.

That the female in the red wig comes up multiple times indicates to me that there is bias in NeuroGen/BigGAN. Is the class vector “female with red wig”? I would imagine it would be more general than that, like “female face” or “human face”. If the latter is the case, it seems almost impossible that different brain areas and subjects would arrive to the same image (albeit with perceptually small differences) if the system were not biased. The woman in the red wig is a prominent example (and may have to do with color?), but other examples (e.g., dog faces) likely exist. Give readers understanding of why this happens and if it is due to bias in the NeuroGen framework or comes from poor prediction of the models (i.e., the bias lies in the linear ensemble).

"The squares in Figure 1 are just illustrations of the weights and the length actually doesn't represent any information about the dimensionality. For clarification, we added the exact number of encoding models used (the value of n) to the figure."

This is misleading to the reader. I think more accurate is to replace in 1b the "Subject 1" "Subject 2" etc. boxes each with a smaller copy of the ResNet50 model in a, and make smaller the "ensemble model" box in b. When I first looked at this figure, I saw " $\text{Beta}_{i,n}$ " and I thought this was going to be a huge matrix—I did not see i and n indexed subjects, making this 8 dimensional. It also seems that each model outputs a vector, not a scalar (because of the lefthand matrix). It is easier to understand if you have one input image, not multiple. I understand these sound like quibbles, but on my first read, it took a lot of time referring to the text to understand Figure 1.

"Figure 2 is not the primary result of the paper but we believe it is important to show the shape of the curve"

Figure 2 does not have any curves. It has an fMRI map (a), bar plots (b), and maximizing images (c). I believe in your rebuttal you are referring to Figure 3; in my review, I am referring to Figure 2. I strongly think Figure 2 could be supplemental or combined with Figure 1 for ease of understanding.

"We decided to keep the violin plots (with median and quartiles) and the data points together"

I would encourage the authors not to use violin plots in the future unless claims are made about the distributions of points (here, only the means are considered to compute p-values, I believe) and if you have more than 100 samples per dimension. Although pretty, a violin plot is an estimate of the probability density of points. Because none of your claims are based on the wiggles of the violin plots, and because you plot all 8 data points, it makes the violin plots redundant (beyond the fact that the distribution curve is doubled due to symmetry). In addition, it opens up questions of how good are your estimates of the density—should you include confidence intervals of the density estimate as well? A simple line for the mean/median makes the most sense here.

Reviewer #3 (Remarks to the Author):

I commend the authors for taking up my previous concerns head-on and for all the additional analyses. The current version of the manuscript does a much better job of backing up the central claims of the paper. I do hope data-sharing is a top priority also given the push from the NIH and other funding bodies. I still do feel that the use of the word 'personalized' is unlike the usage in other fields of science (personalized medicine, personalized genetics, etc.) or commerce (personalized clothing, etc.). But I won't beleaguer this point any further and rather will leave it to the readers to decide. Congratulations to the authors on the study. I recommend publication in its current form

Response to Reviewers

Title: Personalized visual encoding model construction with small data

**Manuscript Reference Number:
COMMSBIO-22-1610A**

Authors:

Zijin Gu

Keith Jamison

Mert Sabuncu

Amy Kuceyeski

Date: November 9, 2022

Message from the Authors

Dear Reviewers,

We thank the reviewers for their recognition of our revised work and the additional comments, which have allowed us to further improve the quality of the manuscript. We have addressed the comments and incorporated these valuable suggestions in the current revision; we believe that the result is a strong manuscript that will be of great interest to the neuroscientific community, vision researchers in particular. The updated contents are colored in blue in the revised manuscript to indicate our changes.

We have made some changes according to the comments, including additional analyses and validation and changes to the wording in the text. All page and figure numbers in our response are based on the revised manuscript, unless otherwise stated. The page and reference numbers mentioned in the reviewers' comments are kept intact and are based on the original manuscript. The references contained in this reviewer response document are in author-year format for ease of reading, and are listed at the end of this document. Thank you and we look forward to hearing the journal's decision.

Sincerely,
Zijin Gu, Keith Jamison, Mert Sabuncu, Amy Kuceyeski

Response To Reviewer #1

Overall Comments

I thank the authors for substantively addressing my comments about introducing noise ceiling estimates. I believe that the results are more interpretable now.

I still believe that the focus on non-linear encoding models is not well motivated scientifically, but this is the authors' paper.

Regarding the response: "we did include the accuracies from models fitted only using NeuroGen data (the NeuroGen's scratch models)." This is still not evident from the main text and figure and should be explicitly stated. At the moment, the NeuroGen scratch model performance is plotted under the title "NeuroGen Subjects (out-of-distribution)", but if I understand the authors' response correctly, this blue violin plot describes within-distribution performance.

Other than that, I have no further comments.

Response

Thank you for your recognition of our revised document. To explicitly state the scratch model was trained using only NeuroGen dataset, we added the below description in the main text:

As there is no large-scale data available for the NeuroGen individuals, we created only the scratch, finetuned, linear ensemble and average ensemble encoding models. The scratch models for NeuroGen individuals were trained using only NeuroGen data, while the latter three were based on the 8 NSD individual-20K models and are thus considered to be out-of-distribution.

We have changed the title in Figure 3 and Figure 4 to be more accurate: "NeuroGen Subjects (out-of-distribution, except scratch)". We also changed the following sentence in the results:

Thus, we did anticipate a drop in the NeuroGen individuals' prediction accuracies compared to the within-distribution NSD individuals' accuracies for the fine-tuned, linear and average ensemble models.

We hope this clarifies the NeuroGen results, and how they relate to the NSD results.

Response To Reviewer #2

Overall Comments

I have re-read my comments and read the authors' rebuttal. I have one major concern that needs to be addressed by the authors in rebuttal. I also have minor concerns that should be addressed by the authors, but I do not need a rebuttal/see a revised manuscript before acceptance.

Response

Thank you for your detailed comments. We hope the below explanations and results could help solve your concern.

Reviewer Major Comment

“We did observe a positive relationship between linear ensemble prediction accuracy and ISC measurement, as shown in the below figure. However, this result is not surprising - if there is a subject that is very similar to another (ISC-measurement is high) then an ensemble model for one of these individuals that includes the other (similar) individual will also be accurate (and vice versa).”

I have a major concern with the metric of consistency, as defined by this manuscript. For real data, it is fine to define ISC consistency as the correlation between responses of two subjects across the same images. This metric can accurately identify the presence of inter-individual differences (but not what those differences are).

However, as displayed by the reviewer figures, using this definition of consistency for the models is redundant with prediction power and leads to ambiguity. I'll give a few examples to make this clear.

First, consider the case where the models (e.g., one linear ensemble model per subject) perfectly predict the subjects' responses. It follows that these models would perfectly match in consistency with the real data (i.e., a large rho for ISC-measurement vs ISC-prediction) because the models perfectly predict all responses. Thus, the models are consistent simply because they have perfect prediction. I suppose this makes sense—perfect prediction should mean that the models perfectly identify inter-individual differences.

Second, consider the case where the responses of the first N-1 subjects are identical but the responses of the Nth subject are uncorrelated (i.e., an ISC=0 between the Nth subject and any other subject). Then consider a model that only outputs responses identical to the responses of the first N-1 subjects. This model would also have perfect consistency, because it too would have an ISC=0 when comparing the model for the Nth subject vs any other subject's model. However, it is unable to predict any response for the Nth subject, so it fails to capture any real inter-individual difference. Its high consistency stems solely from the fact that it has

high prediction.

Third, we now consider the fMRI data in this study. There is a strong correlation between prediction power and consistency (via the reviewer figures). Is it the case that the encoding models are pulling out true inter-individual differences? Or rather this match in consistency (Figure 5 scatter plots) simply arises because the models are better able to predict some subjects more than others? Arguing that prediction is high enough for all subjects (Figure 4) is difficult because the model could be predicting the portions of responses shared across all subjects. A potential confound is that the models do not pull out true inter-individual differences; instead, the match in consistency is because some models fail to predict the “outlier” subjects.

In the end, I think using the Pearson correlation as a consistency metric is too coarse. *The manuscript needs to convince the reader that the models truly pull out inter-individual differences.* To that end, one idea is to analyze the residuals between two true subjects versus the residuals between two models. Wouldn't taking the correlation between these two residual vectors be a better measure of consistency? This metric would still rely on prediction power in the sense that models with perfect prediction would have $\rho=1$. However, it better eliminates the possibility that a correlation in consistency might be caused because a model cannot predict an outlying subject's responses. A high consistency here indicates that the response differences between subjects are correlated with the response differences between models.

Response

We think the consistency metric as defined in the current manuscript is able to assess the models' ability to preserve inter-individual differences. We do acknowledge that there is a somewhat unsurprising relationship between prediction accuracy and consistency, as is described so nicely in your first point - and discussed in the limitations section of the paper.

For the second scenario, as long as we understand the example you described correctly, we respectfully disagree with your conclusion. To make your second example more concrete: let's assume we have 4 subjects and the first 3 have identical measured responses while the last subject has totally uncorrelated measured responses do not correlate at all with compared to the first 3. In this case, the vectorized version of the upper triangular portion of ISC-measurement matrix would be $[1, 1, 0, 1, 0, 0]$. If the encoding model only outputs responses identical to the responses of the first 3 subjects, then a vectorized version of the ISC-prediction matrix would be $[1, 1, 1, 1, 1, 1]$. And the prediction consistency, which is the correlation between the ISC-measurement and ISC-prediction, would not be perfect - it would be undefined or very low (if the model prediction is not exactly identical but instead has some small variations). We hope this concrete example has convinced you of our measure's worth (albeit somewhat influenced by prediction accuracy, as we admit in the discussion).

We offer two additional examples of where our measure of model consistency and prediction accuracy diverge. First, we would like to emphasize the fact that the average ensemble model has similar prediction accuracy to the linear ensemble but has ISC-predictions of all 1 (as they are identical), thus leading to undefined consistency value. We would additionally like to call the reviewer's attention to the results in Supplemental Figure S5, which is copied below. There, we show that the linear ensemble's accuracy approaches its maximum value (for the model including all $N = 8$ subjects) when only using $N = 2$ and $N = 1$ subjects' individual-20K models, but that model

consistency continues to increase as more and more subjects' individual-20k models are added to the linear ensemble predictions. These two examples are contrary to the idea that model accuracy and model consistency are entirely redundant.

If we do indeed proceed as suggested and take the residual (l_2 distance) between two subjects' measured and predicted responses and correlate them, the resulting metric will be identical to the current consistency metric. To explain: assume we have two subjects, subject 1 and 2, and their predicted (or measured) responses are y_1 and y_2 , respectively. The Pearson correlation and the l_2 distance between y_1 and y_2 would then be:

$$r = \frac{\sum_i (y_{1_i} - \mu_1)(y_{2_i} - \mu_2)}{\sqrt{\sum_i (y_{1_i} - \mu_1)^2 \sum_i (y_{2_i} - \mu_2)^2}} \quad \text{and} \quad MSE = \frac{1}{n} \sum_i (y_{1_i} - y_{2_i})^2$$

Different subjects have different magnitude of fMRI response values (and thus also predicted responses that are fit using MSE to the fMRI responses) due to various experimental, physiological and computational factors so subjects' measured and predicted values first need to be normalized before calculating the residual. This means y_1 and y_2 have mean 0 and variance 1, and thus we have $\frac{1}{n} \sum_i y_{1_i}^2 = 1$ and $\frac{1}{n} \sum_i y_{2_i}^2 = 1$, which reduce the correlation above to $r = \frac{1}{n} \sum_i y_{1_i} y_{2_i}$ and MSE to $MSE = 2(1 - \frac{1}{n} \sum_i y_{1_i} y_{2_i}) = 2(1 - r)$. Because the MSE and correlation are proportional, taking the correlation between two ISCs and two l_2 residuals will result in the same consistency value.

The above explanations make the case for the utility of the prediction consistency metric, and, coupled with our discussion of the potential overlap between model accuracy and consistency in the limitations section, we believe makes the case for using this metric. We have added the following text to the discussion

Across individuals, consistency and model accuracy are somewhat entwined. If a query subject's responses are very similar to the reference individuals' responses (and thus have high ISC-measurement and high ISC-prediction), the accuracy of the ensemble model will also be high. On the other hand, if a query subject's responses are uncorrelated with the reference individuals, the ISC-measurement and accuracy will be low but the ISC-prediction will remain high (as it will be based on the reference individual's models that are all very similar to one another). This is one consideration to make when interpreting the consistency metric. However, they are clearly not entirely overlapping, as evidenced by two facts: that the average ensemble has similar accuracy compared to the linear ensemble but undefined consistency and Supplementary Figure S5 which demonstrates that maximal linear ensemble model accuracy is achieved with only 1 or 2 reference subjects while model consistency continues to increase as more and more individuals are added to the model.

Reviewer Minor Comment

“We agree with your thoughts that perhaps the ensemble models perform as well as the individual-20K models is because they don't need to fit such a large number of parameters. However, we believe it is that the individual-20K models are not overfitted”

I apologize, I was using the broad definition of the term “overfitting”. I did not mean that the particular individual-20K models were overfitting due to the optimization procedure or regularization (which you make clear in the training curve figure). Instead, another form of “regularization” is the number of parameters that need to

Figure S5: Relationship between number of pretrained encoding models included in the linear ensemble approach and **a,b,c,d** prediction accuracy and **e** prediction consistency in NSD subjects; **f,g,h,i** prediction accuracy and **j** prediction consistency in NeuroGen subjects.

be fit. I'm thinking about the case of fitting a linear line vs a polynomial fit with a large degree—the latter will “overfit” if it does not have enough data. This is one potential cause why the individual-20K model does not achieve better performance. Regardless, mention to the reader why performance is about the same for the individual-20K vs the average ensemble model (Fig. 3)—your reasoning about hitting the upper bound of prediction makes sense, as well as concerns for overfitting. It is an interesting point to bring up, because one could argue the ensemble itself is prone to overfitting (as it relies on 7 large models). This might be confusing to readers of why this works—it would be helpful to clarify.

Response

We have added the following text to the discussion section:

Relatedly, we found that the linear/average ensemble models perform as well as the individual-20k models for the NSD data, which could either be explained by the individual-20K overfitting (although not likely from our assessments of model fit) or the fact that all three models are approaching the ceiling of possible accuracy determined by the relatively low SNR of the fMRI data.

Reviewer Minor Comment

“The images of an individual with red hair are generated using the same class vector (across different regions/subjects) but have different noise vectors, therefore they look similar but are not identical.”

The part that tripped me in Figure 6 caption is “The image generator takes in a class vector which is fixed during optimization. . .”. Change this language to indicate that different images may have different class vectors.

That the female in the red wig comes up multiple times indicates to me that there is bias in NeuroGen/BigGAN. Is the class vector “female with red wig”? I would imagine it would be more general than that, like “female face” or “human face”. If the latter is the case, it seems almost impossible that different brain areas and subjects would arrive to the same image (albeit with perceptually small differences) if the system were not biased. The woman in the red wig is a prominent example (and may have to do with color?), but other examples (e.g., dog faces) likely exist. Give readers understanding of why this happens and if it is due to bias in the NeuroGen framework or comes from poor prediction of the models (i.e., the bias lies in the linear ensemble).

Response

We changed the caption of Figure 6 to the following:

One synthetic image is generated from a 1000 dimension, one-hot encoded class vector (corresponding to one class in ImageNet) and a noise vector which will be identified through optimization. Different synthetic images may have different class vectors.

We have added the following text to the results:

Additionally, BigGAN has a truncation parameter that allows generation of sets of images with varying balance of variety and fidelity. If the truncation parameter is large, the images will have more variety but look less realistic than if the truncation parameter is small. Our images were generated using truncation parameter 0.1 in order to enforce more realistic images, of course at the cost of decreased variability within a category.

Here have attached some examples of images generated from the "wig" category (the category that was discussed in your comment above) using a large truncation parameter. You can indeed see more variety but also less fidelity - in fact, one image appears to be nonsensical.

Figure: Example synthetic images from "wig" class with truncation parameter 0.8.

Reviewer Minor Comment

“The squares in Figure 1 are just illustrations of the weights and the length actually doesn’t represent any information about the dimensionality. For clarification, we added the exact number of encoding models used (the value of n) to the figure.”

This is misleading to the reader. I think more accurate is to replace in 1b the “Subject 1” “Subject 2” etc. boxes each with a smaller copy of the ResNet50 model in a, and make smaller the “ensemble model” box in b. When I first looked at this figure, I saw “Beta _{i,n} ” and I thought this was going to be a huge matrix—I did not see i and n indexed subjects, making this 8 dimensional. It also seems that each model outputs a vector, not a scalar (because of the lefthand matrix). It is easier to understand if you have one input image, not multiple. I understand these sound like quibbles, but on my first read, it took a lot of time referring to the text to understand Figure 1.

Response

Thank you for your detailed suggestion for improving the figure - see the new figure below.

Reviewer Minor Comment

“Figure 2 is not the primary result of the paper but we believe it is important to show the shape of the curve”

Figure 2 does not have any curves. It has an fMRI map (a), bar plots (b), and maximizing images (c). I believe in your rebuttal you are referring to Figure 3; in my review, I am referring to Figure 2. I strongly think Figure 2 could be supplemental

a Encoding model architecture

b Ensemble model architecture

Figure 1: **a** The encoding model architecture. A feature extractor adopted from ResNet-50 ? extracts features from the input image and a linear readout maps the extracted features to the responses for a specific brain region. **b** The ensemble model architecture. A group of n pretrained encoding models (where $n = 7$ or 8) are used to obtain a set of predicted activations, which are combined via a linear model fitted via ordinary least squares to optimally predict the query subject's regional brain response.

or combined with Figure 1 for ease of understanding.

Response

We have now moved Figure 2 to supplementary material.

Reviewer Minor Comment

“We decided to keep the violin plots (with median and quartiles) and the data points together”

I would encourage the authors not to use violin plots in the future unless claims are made about the distributions of points (here, only the means are considered to compute p-values, I believe) and if you have more than 100 samples per dimension. Although pretty, a violin plot is an estimate of the probability density of points. Because none of your claims are based on the wiggles of the violin plots, and because

you plot all 8 data points, it makes the violin plots redundant (beyond the fact that the distribution curve is doubled due to symmetry). In addition, it opens up questions of how good are your estimates of the density—should you include confidence intervals of the density estimate as well? A simple line for the mean/median makes the most sense here.

Response

We agree with the reviewer that the violin plot may be misleading, so we have replaced them with a cloud plot of all the points and a line for the group mean.

Response To Reviewer #3

Overall Comments

I commend the authors for taking up my previous concerns head-on and for all the additional analyses. The current version of the manuscript does a much better job of backing up the central claims of the paper. I do hope data-sharing is a top priority also given the push from the NIH and other funding bodies. I still do feel that the use of the word ‘personalized’ is unlike the usage in other fields of science (personalized medicine, personalized genetics, etc.) or commerce (personalized clothing, etc.). But I won’t beleaguer this point any further and rather will leave it to the readers to decide. Congratulations to the authors on the study. I recommend publication in its current form.

Response

Thank you for your recognition of our revised document. We thank you for all the constructive comments which helped make the paper stronger.

REVIEWERS' COMMENTS:

Reviewer #3 (Remarks to the Author):

Thank you for your rebuttal. I have now corrected my comments to better reflect my opinion. I think the defined ISC metric is not particularly helpful; I am worried that another manuscript will come along soon refuting this ISC metric, at the detriment of this paper. Thus, I suggest a more helpful metric. I encourage the authors to either adopt another metric or defend the ISC metric with simulations in a supplemental figure to make it clear how ISC and prediction power are related (including a simulated example in which they are independent)---and what conclusions we can make with prediction power and ISC.

Comments:

Apologies and please let me update my second case, as I missed a crucial detail. The key difference between this case and my previous "second case" is that the Nth model outputs noise: Consider N subjects. The first N-1 subjects all have the same responses, but the responses of the Nth subject are uncorrelated with responses of the other N-1 subjects. Now consider the corresponding N models. Let the first N-1 models perfectly predict the responses of the first N-1 subjects. However, the Nth model is a poor fit and outputs noise. Thus, the Nth model fails to predict the responses of the Nth subject *and* its responses are uncorrelated with responses to any of the other N-1 models. In this case, the ISC between measurements and predictions is 1 (e.g., for N=3, the upper triangle is [1,1,0,1,0,0] for both models and real data). However, this is purely due to changes in prediction power...the Nth model fails at capturing any meaningful difference between the Nth subject and the other N-1 subjects. Thus, the defined ISC measurement is unable to tell whether models are able to capture inter-subject differences or whether the differences are simply due to prediction power.

re: average vs. linear ensemble

It still could be the case that the linear ensemble better predicts some subjects more than others. Given that there is a correlation between ISC and prediction power, you have not ruled out the possibility that the weak ISC of the linear ensemble ($r=0.33$, Fig.4e) is more due to prediction power than to actually identifying meaningful differences between subjects.

re: plateauing prediction but increasing consistency

At first glance, this does seem to support the conclusion that consistency and prediction power are pulling out different signals. However, at the end of the day, if your models are able to better pick out differences between subjects, shouldn't their prediction accuracy also increase? If prediction accuracy does not increase, does it suggest that the linear model is picking out inter-subject differences at the cost of failing to predict shared signals between subjects? I think a more parsimonious explanation is that with more pretrained models, the outputs can differ more across subjects; some of those outputs may be overfitting to the training data. To check this, perhaps you can see if the correlation between averaged prediction power and model/measured ISC becomes stronger as you increase the number of pretrained models. If it does, it indicates that the worse-predicting models are becoming more different from the other models but not accurately identifying differences across subjects.

re: using a correlation between residual vectors

Again, apologies, I did not think you would interpret my proposed metric in that way---my fault! I do not mean to take the L2/correlation between subject 1 and 2; the authors are correct that this is basically the same as the correlation.

Instead, here is what I meant:

Consider response vectors y_i and y_j from the i th and j th subjects, respectively, where y_k is a vector of length N for N images. Then, consider model prediction vectors \hat{y}_i and \hat{y}_j , both of length N. Take the measured residual vector $v_{ij} = y_i - y_j$. Take the prediction residual vector $\hat{v}_{ij} = \hat{y}_i - \hat{y}_j$. Now, compute the correlation R_{ij} between vectors v_{ij} and \hat{v}_{ij} . Take as our metric this correlation averaged over pairs of subjects (the upper triangle of R_{ij}). This tells us that the inter-subject differences between subjects are similar or not between

measured and predicted vectors. It directly measures if the model residuals between subjects i and j match the measured residuals between subjects i and j . Thus, going back to the "second case", this residual correlation would be lower because the N th model's residuals do not match the residuals for the N th subject.

Response to Reviewers

Title: Personalized visual encoding model construction with small data

**Manuscript Reference Number:
COMMSBIO-22-1610B**

Authors:

Zijin Gu

Keith Jamison

Mert Sabuncu

Amy Kuceyeski

Date: November 28, 2022

Response To Reviewer #3

Overall Comments

Thank you for your rebuttal. I have now corrected my comments to better reflect my opinion. I think the defined ISC metric is not particularly helpful; I am worried that another manuscript will come along soon refuting this ISC metric, at the detriment of this paper. Thus, I suggest a more helpful metric. I encourage the authors to either adopt another metric or defend the ISC metric with simulations in a supplemental figure to make it clear how ISC and prediction power are related (including a simulated example in which they are independent)—and what conclusions we can make with prediction power and ISC.

Response

Thank you for correcting your comments. We hope the below explanations and results could help solve your concern.

Reviewer Comment

Apologies and please let me update my second case, as I missed a crucial detail. The key difference between this case and my previous "second case" is that the Nth model outputs noise: Consider N subjects. The first N-1 subjects all have the same responses, but the responses of the Nth subject are uncorrelated with responses of the other N-1 subjects. Now consider the corresponding N models. Let the first N-1 models perfectly predict the responses of the first N-1 subjects. However, the Nth model is a poor fit and outputs noise. Thus, the Nth model fails to predict the responses of the Nth subject *and* its responses are uncorrelated with responses to any of the other N-1 models. In this case, the ISC between measurements and predictions is 1 (e.g., for N=3, the upper triangle is [1,1,0,1,0,0] for both models and real data). However, this is purely due to changes in prediction power...the Nth model fails at capturing any meaningful difference between the Nth subject and the other N-1 subjects. Thus, the defined ISC measurement is unable to tell whether models are able to capture inter-subject differences or whether the differences are simply due to prediction power.

Response

We thank you for the clarification of your example; however, we still do not agree with your conclusion based on your clarification of the example provided. If the Nth subject's responses are uncorrelated with responses of the other N-1 subjects, and the Nth model's predictions are also uncorrelated with the other N-1 subjects' model predictions, then the models' predictions have indeed reflected the true underlying inter-subject variability in the responses. Whether a model is predicting something uncorrelated or correlated with measurement is prediction accuracy (which we also quantify and report in the manuscript), not consistency.

Reviewer Comment

re: average vs. linear ensemble It still could be the case that the linear ensemble better predicts some subjects more than others. Given that there is a correlation between ISC and prediction power, you have not ruled out the possibility that the weak ISC of the linear ensemble ($r=0.33$, Fig.4e) is more due to prediction power than to actually identifying meaningful differences between subjects.

Response

We would like to clarify that the correlation mentioned ($r=0.33$, Fig.4e) is not ISC but prediction consistency (which is the correlation between ISC-prediction and ISC-measurement). If the above comment is referring to the ISC-prediction values, they are between 0.7-0.95 for the linear ensemble model, which is higher than the scratch (0.1-0.8) and fine-tuned (0.4-0.85) models and similar to the individual-20K model (0.55-0.9). If the above comment is in fact referring to a possible correspondence between model consistency and accuracy, then we refer the reviewer to the fact that the individual 20K model and the linear ensemble model have the best accuracy values (higher than scratch and fine-tuned models, see Figure 3) but that the scratch and fine tuned models have better consistency compared to the linear ensemble and worse consistency compared to the individual-20K model (Figure 4). This fact is in addition to several other items of supporting evidence in our current and previous responses that, while somewhat inter-related (as we discuss in the limitations section of the paper), consistency (or ISC-measurement) does not equal accuracy.

Reviewer Comment

re: plateauing prediction but increasing consistency At first glance, this does seem to support the conclusion that consistency and prediction power are pulling out different signals. However, at the end of the day, if your models are able to better pick out differences between subjects, shouldn't their prediction accuracy also increase? If prediction accuracy does not increase, does it suggest that the linear model is picking out inter-subject differences at the cost of failing to predict shared signals between subjects? I think a more parsimonious explanation is that with more pretrained models, the outputs can differ more across subjects; some of those outputs may be overfitting to the training data. To check this, perhaps you can see if the correlation between averaged prediction power and model/measured ISC becomes stronger as you increase the number of pretrained models. If it does, it indicates that the worse-predicting models are becoming more different from the other models but not accurately identifying differences across subjects.

Response

As we discussed in the paper, we believe model accuracy is hitting a noise ceiling that arises from the relatively noisy measured responses. So, prediction accuracy won't continue to increase even if the model has increasing ability in picking out differences between subjects. To belabor this point

further, we did the experiment suggested and below plot the correlation between prediction accuracy and ISC-measurement with increasing number of pretrained models: It seems from the figure that

Figure: Change of the correlation between prediction accuracy and ISC-measurement as increasing the number of pretrained model.

the correlation between prediction accuracy and ISC-measurement remains almost identical as we increase the number of pretrained models for all ROIs in NSD, while for NeuroGen there are more fluctuations but doesn't show a consistent pattern of increase with increasing pretrained models. This contradicts the idea that the worse-predicting models are accurately identifying differences across subjects instead of just becoming more different from the other models as the number of pretrained models increases.

Reviewer Comment

re: using a correlation between residual vectors Again, apologies, I did not think you would interpret my proposed metric in that way—my fault! I do not mean to take the L2/correlation between subject 1 and 2; the authors are correct that this is basically the same as the correlation.

Instead, here is what I meant: Consider response vectors y_i and y_j from the i th and j th subjects, respectively, where y_k is a vector of length N for N images. Then, consider model prediction vectors \hat{y}_i and \hat{y}_j , both of length N . Take the measured residual vector $v_{ij} = y_i - y_j$. Take the prediction residual vector $\hat{v}_{ij} = \hat{y}_i - \hat{y}_j$. Now, compute the correlation R_{ij} between vectors v_{ij} and \hat{v}_{ij} . Take as our metric this correlation averaged over pairs of subjects (the upper triangle of R_{ij}). This tells us that the inter-subject differences between subjects are similar or not between measured and predicted vectors. It directly measures if the model residuals between subjects i and j match the measured residuals between subjects i and j . Thus, going

back to the "second case", this residual correlation would be lower because the Nth model's residuals do not match the residuals for the Nth subject.

Response

Thank you for clarifying your proposed metric. We have calculated the proposed metric for further exploration of consistency and the result is that the models exhibit similar relative values compared to one another. The upper triangular part of the individual-20K model's matrices R (as defined above), across the four regions of interest was 0.14. In a similar pattern to the consistency metric in the paper, the values of the scratch and fine-tuned models were about 40-65% of the individual-20k model's value and the value of the linear ensemble model was about 55% of the individual-20K model's value. Based on all the provided analysis, we believe we have sufficient evidence that the current consistency metric is able to capture the model's ability in preserving inter-subject differences.